# Mitochondrial acetyl-CoA reversibly regulates locus-specific histone acetylation and gene expression

Oswaldo A Lozoya[1], Tianyuan Wang[1], Dagoberto Grenet[1], Taylor C Wolfgang[1], Mack Sobhany[1], Douglas Ganini da Silva[2], Gonzalo Riadi[3], Navdeep Chandel[4], Richard P Woychik[1] , Janine H Santos[1]

The impact of mitochondrial dysfunction in epigenetics is emerging, but our understanding of this relationship and its effect on gene expression remains incomplete. We previously showed that acute mitochondrial DNA (mtDNA) loss leads to histone hypoacetylation. It remains to be defined if these changes are maintained when mitochondrial dysfunction is chronic and if they alter gene expression. To fill these gaps of knowledge, we here studied a progressive and a chronic model of mtDNA depletion using biochemical, pharmacological, genomics, and genetic assays. We show that histones are primarily hypoacetylated in both models. We link these effects to decreased histone acetyltransferase activity unrelated to changes in ATP citrate lyase, acetyl coenzyme A synthetase 2, or pyruvate dehydrogenase activities, which can be reversibly modulated by altering the mitochondrial pool of acetyl-coenzyme A. Also, we determined that the accompanying changes in histone acetylation regulate locus-specific gene expression and physiological outcomes, including the production of prostaglandins. These results may be relevant to the pathophysiology of mtDNA depletion syndromes and to understanding the effects of environmental agents that lead to physical or functional mtDNA loss.

## Introduction

The role of mitochondria in cell biology and organismal health has expanded dramatically in the last decade. From a focus originally on bioenergetics, it is now recognized that mitochondria broadly affect cell physiology in diverse ways. For instance, mitochondria interact with other organelles, such as the endoplasmic reticulum, by close contacts or through the generation of small vesicular carriers, which allows the transport and exchange of lipids, proteins and other small molecules such as calcium (Csordas et al, 2010; Sugiura et al, 2014). Mitochondria are also important players in

signaling via reactive oxygen species and other metabolites that impart posttranslational modifications to many proteins, including transcription factors (Chandel, 2015). Most recently, we and others have shown that mitochondria influence the epigenome (Smiraglia et al, 2008; Martinez-Reyes et al, 2016; Liu et al, 2017; Lozoya et al, 2018), yet full mechanistic insights and outcomes of this relationship are still lacking.

The relevance of better understanding the impact of mitochondrial function in epigenetics cannot be understated, given the many ways mitochondrial output has been documented to influence gene expression (Durieux et al, 2011; Gomes et al, 2013; Picard et al, 2014). Novel links between mitochondrial function and epigenetics continue to be unveiled and mechanistic understanding of this relationship is emerging. Tricarboxylic acid (TCA) cycle intermediates such as acetyl-coenzyme A (CoA) and $\alpha$-ketoglutarate ($\alpha$-KG) are substrates or cofactors for enzymes that alter the epigenome, such as the histone acetyltransferases (HATs) and the demethylases (Smiraglia et al, 2008; Wallace, 2009; Minocherhomji et al, 2012; Meyer et al, 2013). Thus, mitochondrial dysfunction could, for example, alter the nuclear epigenome through reduced TCA flux. In fact, we first reported that progressive loss of mitochondrial DNA (mtDNA) and the associated changes in TCA cycle output, by ectopically expressing a dominant-negative mtDNA polymerase (DN-POLG), led to histone hypoacetylation in the nucleus (Martinez-Reyes et al, 2016). Using this same cell system, we also demonstrated a direct link between loss of mtDNA and DNA hypermethylation, which we showed was driven by modulation of methionine salvage and polyamine synthesis, both sensitive to changes in TCA cycle flux. We showed that DNA methylation changes occurred predominantly at the promoters of genes that responded to mitochondrial dysfunction, increased progressively over the course of mtDNA depletion, and could be reversed by maintaining NADH oxidation in the mitochondria, even in the context of complete mtDNA loss (Lozoya et al, 2018).

Although our initial work using the DN-POLG system revealed hypoacetylation of histones in the nucleus as a function of

---

[1]Genome Integrity and Structural Biology Laboratory, National Institute of Environmental Health Sciences, National Institutes of Health, Durham, NC, USA  [2]Immunity, Inflammation and Disease Laboratory, National Institute of Environmental Health Sciences, National Institutes of Health, Durham, NC, USA  [3]Centro de Bioinformática y Simulación Molecular, Facultad de Ingeniería, Universidad de Talca, Talca, Chile  [4]Department of Medicine, Northwestern University Feinberg School of Medicine, Chicago, IL

Correspondence: janine.santos@nih.gov; rick.woychik@nih.gov

progressive mtDNA loss (Martinez-Reyes et al, 2016), mechanistic details associated with these effects were not interrogated. Importantly, it remains unknown whether those histone changes are sufficient to alter gene expression and impact functional outcomes. In this work, we used the DN-POLG cells together with a model of chronic mtDNA depletion to establish cause–effect relationships. Using several biochemical, transcriptomics, epigenomics, genetics, and pharmacological approaches, we found that histone acetylation loss or gain occurred predominantly on the promoters of differentially expressed genes (DEGs), that even chronic transcriptomic changes were amenable to inducible epigenetic manipulation by supplementation with TCA cycle intermediates, and that altered histone acetylation status largely preceded gene expression remodeling.

## Results

### Changes in H3K9ac levels by progressive mtDNA depletion occurs early in the course of mtDNA loss and predominantly in the promoters of DEGs

Using Western blots and quantitative mass spectrometry, we previously determined that progressive mtDNA depletion in the DN-POLG cells led to histone acetylation changes at specific lysine residues on H3, H2B, and H4; H3 acetylation changes were more frequent and pronounced (Martinez-Reyes et al, 2016). In addition, we also unexpectedly found that lysine acetylation increased in some histones (Martinez-Reyes et al, 2016). Most recently, we described that loss of mtDNA in this cell model was accompanied by progressive transcriptional remodeling (Lozoya et al, 2018), providing an excellent platform to interrogate the extent to which the histone acetylation changes were involved in regulating the expression of those genes. To address this question, we performed chromatin immunoprecipitation followed by deep sequencing (ChIP-seq) in the DN-POLG cells at days 0, 3, 6, and 9. We studied H3K9ac enrichments for several reasons, including the fact that this is primarily a promoter mark and that it was decreased about ~50% at day 9 in the DN-POLG cells (Martinez-Reyes et al, 2016).

We started by examining the relative *de novo* H3K9ac peak enrichment around the transcriptional start site (TSS) of genes at days 0, 3, 6, or 9 as recommended elsewhere (Nakato & Shirahige, 2017). For more details, see the Materials and Methods section. Using this approach, we found progressive loss of average genome-wide H3K9ac peak densities over time. Notably, the changes in H3K9ac enrichment were significant already at day 3 (Fig 1A, blue lines), which was unexpected, given that neither Western blots nor mass spectrometry had shown significant changes at this time (Martinez-Reyes et al, 2016). However, these data underscore the sensitivity of ChIP-seq compared with these other approaches for analysis of histone modification abundance. Based on the total number of reproducible peaks and visual inspection of the genomic tracks at each time point, it became clear that peaks were either lost or significantly decreased in the DN-POLG samples over time (Fig S1A).

We next identified the peaks that showed statistical differences at days 3, 6, or 9 relative to day 0 and performed hierarchical clustering to define how they behaved over time. We found that most peaks decreased as mtDNA was depleted (Fig 1B, cluster A), whereas a few increased between days 3 and 6 with levels at day 9 resembling those of day 0 (Fig 1B, cluster B). Detailed analysis of the peaks following the behavior of cluster B indicated that, in all cases, the peak at the canonical annotated TSS decreased. However, a new peak close to the TSS, but in a non-annotated region of the gene, emerged and was the one that increased (see example on Fig S1B); the relevance and origin of these novel peaks remain unclear. We

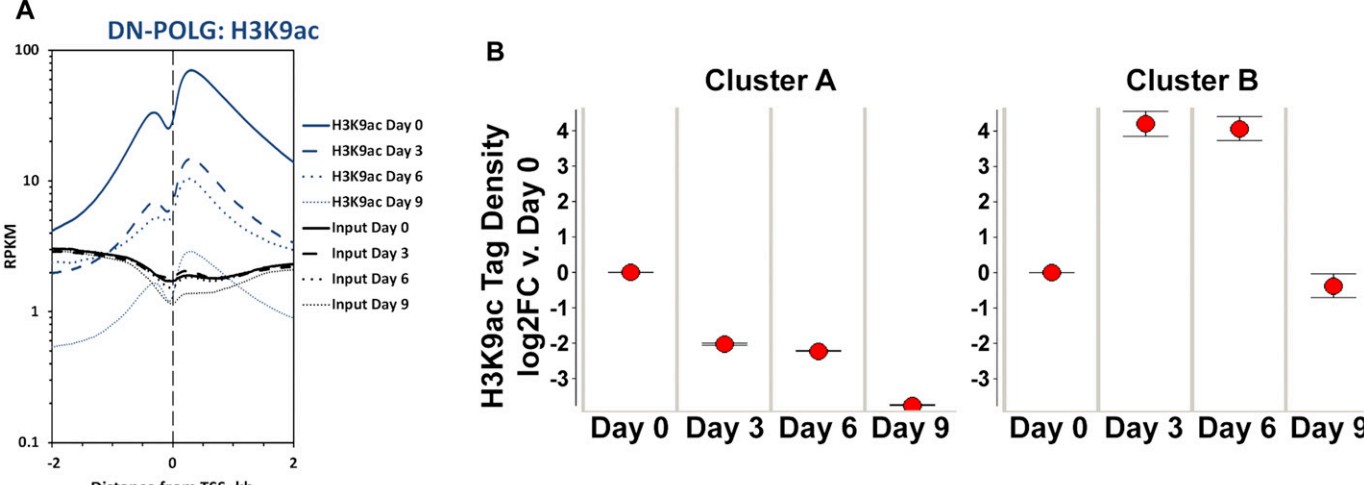

**Figure 1. Changes in histone acetylation caused by progressive mtDNA depletion occur in specific loci and are robust already at day 3.**
**(A)** Average genome-wide enrichment level of H3K9ac peaks DNA centered around the TSS of genes. N = 2 per time point. Blue line shows data for H3K9ac and black lines for input DNA. **(B)** Time-dependent patterns of average significantly different enriched H3K9ac peaks relative to day 9 in the DN-POLG cells following hierarchical clustering; error bars represent standard error of the mean.

then cross-referenced the coordinates of the significantly changed peaks with those of the promoters of the 2,854 DEGs identified on the DN-POLG based on RNA-seq (Lozoya et al, 2018). We found that ~73% of DEGs in the DN-POLG cells harbored a change in their promoter H3K9ac peak levels (Table S1). The promoters of 1,995 followed the pattern shown in cluster A, whereas 65 DEGs followed the pattern of cluster B (Table S1). Although we found that modulation in histone peak intensities were detected in ~10,000 genes (Fig S1C, and Table S2), including those transcribed but not differentially expressed, statistical analysis revealed that the odds ratio of a gene being differentially expressed and having a change in its H3K9ac level was OR = 2.41; $P < 5 \times 10^{-90}$ (Fig S1C). These data indicate that modulation of H3K9ac was more prevalent in the promoters of DEGs than of other transcribed genes. The pathways enriched by genes harboring altered H3K9ac levels are shown in Table S1 and revealed that they were involved in key functions affected in our model system. For example, we had previously shown that cholesterol biosynthesis, putrescine metabolism, and the TCA cycle, among other pathways, were inhibited by progressive mtDNA depletion (Lozoya et al, 2018). Concomitant to the decreased expression of genes in those pathways was the loss of their H3K9ac promoter mark (Table S1). Taken together, these data suggest that the mitochondria-driven histone acetylation changes strongly correlate with the differential expression of genes that respond to mtDNA depletion.

### Maintenance of site-specific H3K9ac levels coincides with loss of differential expression of genes and reversal of functional outcomes caused by progressive mtDNA depletion

Resuming NADH oxidation in the mitochondria of the DN-POLG cells through the ectopic expression of NADH dehydrogenase like I (NDI1) and alternative oxidase (AOX) maintained histone acetylation levels even when mtDNA was completely lost (Martinez-Reyes et al, 2016). Concomitantly, only 23 genes were shown to be differentially expressed in these cells between days 0 and 9 by microarrays, whereas approximately 1,000 genes were identified in the DN-POLG at this same time using this tool (Lozoya et al, 2018). Thus, if histone acetylation changes were required for the differential gene expression, then the promoter H3K9ac levels in the coordinates of the 1,000 genes identified in the DN-POLG at day 9 should be changed. Conversely, H3K9ac enrichment in those same loci should be maintained across time in cells expressing NDI1/AOX. To test this hypothesis, we performed ChIP-seq in the NDI1/AOX-expressing cells following the same procedures used for the DN-POLG. We found that the average genome-wide H3K9ac peak densities were not changed when cells expressed NDI1/AOX, even when mtDNA was completely lost at day 9 (Fig 2A). Focusing solely on the promoter coordinates of the ~1,000 DEGs, we found that average H3K9ac levels were significantly increased in the NDI1/AOX cells, irrespective of whether peaks followed the pattern of cluster A or B compared with the levels found in the DN-POLG cells (Fig 2B). Analysis of individual promoters further confirmed that H3K9ac enrichments in the same genomic coordinates were increased in the NDI1/AOX cells compared with the densities identified in the DN-POLG counterparts (representative data on Fig 2C). It is noteworthy that the average fold-change in the expression of genes in

those coordinates in the NDI1/AOX–expressing cells at day 9 were minor and not statistically significant, following the H3K9ac peak densities (Fig 2D). Thus, we conclude that there is a strong correlation between modulation of the H3K9ac promoter levels and differential gene expression (or lack thereof) in response to mitochondrial dysfunction.

The existence of metabolomics data for the DN-POLG and NDI1/AOX cells (Lozoya et al, 2018) provided us with a unique opportunity to determine whether the gene expression changes associated with H3K9ac peak densities had functional outcomes. Many TCA cycle genes had decreased H3K9ac levels and showed inhibited gene expression and decreased metabolite levels in the DN-POLG cells, all of which were rescued in the NDI1/AOX-expressing cells (Table S1 and [Lozoya et al, 2018]). We found that, likewise, genes associated with pathways not directly impacted by resumption of mitochondrial NADH oxidation, such as cholesterol biosynthesis, also showed decreased H3K9ac levels (Table S1). All of the genes associated with cholesterol biosynthesis captured in our RNA-seq analyses were down-regulated in the DN-POLG cells (Fig 2E). Only one metabolite directly reflecting cholesterol biosynthesis, lathosterol, was found in our metabolomics analysis (Lozoya et al, 2018), but its levels were decreased in the DN-POLG cells and were completely rescued in the NDI1/AOX cells (Fig 2F). Similarly, other metabolites associated with pathways that could impact cholesterol synthesis were completely rescued in the NDI1/AOX-expressing cells (Fig 2G). Collectively, these results support the conclusion that the effects of mtDNA depletion on histone acetylation led to functional outcomes, including both at the level of gene expression and with respect to the metabolic products of the genes/pathways affected.

### Histones are hypoacetylated under chronic mitochondrial dysfunction and are associated with decreased HAT activity

Although the genetic manipulation performed in the DN-POLG and NDI1/AOX cells provided unequivocal inferences about the role of mitochondria in epigenetically driven gene expression regulation, a drawback of these models is that the histone and transcriptome rescues were performed in isogenic but independent cell lines. Ideally, the modulation of the histone mark and downstream effects should be demonstrated in the same cellular background. Hence, we next performed a series of experiments in an unrelated cell line (143B) whose mtDNA had been depleted by exposure to low doses of ethidium bromide (EtBr) (King & Attardi, 1989); herein, these cells are referred to as rho0 and the mtDNA-repleted control counterpart as rho+. Experiments included biochemical evaluation of parameters associated with mitochondrial function (Fig S2A–D) and the demonstration that 143B rho0 cells have H3K9ac and H3K27ac marks chronically depleted when compared with rho+ controls as judged by Western blots (Fig 3A). Overall, these results recapitulate the findings on the DN-POLG model. Also, they suggest the lack of an active compensatory mechanism to maintain global histone acetylation when mtDNA is depleted—whether short- or long term.

To modulate the histone marks in the 143B rho0, it was first necessary to gain insights into the mechanisms associated with their hypoacetylation phenotype. We started out by monitoring

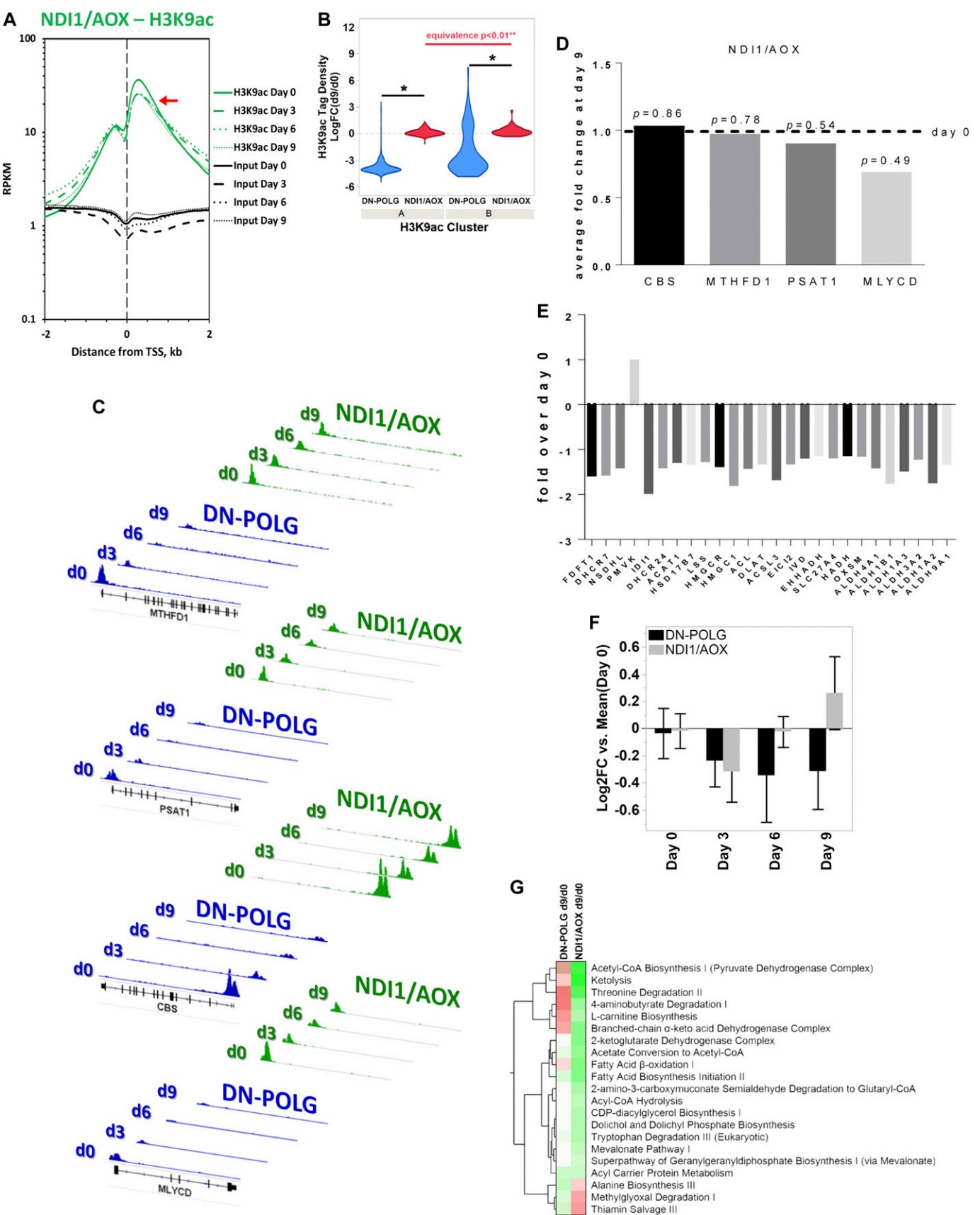

global histone deacetylase (HDAC) and acetyl–transferase activities given that these are opposing functions that could affect the steady state level of histone acetylation in cells. Although no differences were observed in HDAC activity between rho+ and rho0 cells (Fig 3B), which ruled out a role for increased deacetylation of histones, HAT activity was ~50% reduced in rho0 cells (Fig 3C). The influence of decreased mtDNA content on HAT enzymatic function was further confirmed in 143B cells freshly depleted of mtDNA by EtBr exposure (Fig S3A–C) and in mouse embryonic fibroblasts from the TFAM (transcription factor A mitochondria) heterozygote mouse (Fig S3D), which shows 50% reduction in the amount of mtDNA (Fig S3E and [Woo et al, 2012]).

Next, we estimated the levels of cellular acetyl-CoA because this metabolite is not only the substrate for HAT function but also, in mammalian cells, it is primarily generated from mitochondrial-derived citrate by cytosolic ATP citrate lyase (ACL). Estimation of total levels of acetyl-CoA using a fluorescent-based assay confirmed significantly decreased levels of acetyl-CoA in rho0 cells compared with rho+ cells (Fig3D). Given inhibition of ACL was previously shown to decrease acetyl-CoA pools leading to histone hypoacetylation in the nucleus (Wellen et al, 2009), decreased ACL function could mediate the effects of mtDNA depletion on HAT activity and histone acetylation. However, neither ACL protein content nor enzymatic activity, as judged by phosphorylation at serine 455 (Potapova et al, 2000), was different between rho+ and rho0 cells (Fig 3E). Likewise, acetyl coenzyme A synthetase 2 (ACSS2) and pyruvate dehydrogenase (PDH) nuclear translocation have been proposed to regulate local acetyl-CoA production and histone acetylation levels (Sutendra et al, 2014; Mews et al, 2017; Bulusu et al., 2017). However, no differences in the protein amounts or subcellular distribution were identified for either ACCS2 or PDH under our experimental conditions (Fig 3F). Thus, we conclude that depletion of mtDNA negatively influences HAT activity through chronic decreases in total levels of acetyl-CoA that are not related to impaired ACL, ACSS2, or PDH activities.

Because the 143B rho0 cells were generated by chronic low dose exposure to EtBr, a mutagen, it was formally possible that this treatment mutated HAT genes that impaired their function. Therefore, we compared deep-sequenced nuclear DNA from the rho+ or rho0 cells to the reference genome and showed similar changes in both cells (Fig S4A), ruling out that an increased mutation burden impaired HAT function in rho0 cells. Likewise, no significant decreases in the transcription of HAT or acetyl-transferase genes were identified in the rho0 compared with the rho+ by microarrays (Table S3). Protein amounts identified for two main HATs, GCN5, and EP300 (Fig S4B), which acetylate K9 and 27 residues in histone 3 tails (Creyghton et al, 2010; Zhang et al, 2015), were also not different between rho+ and rho0.

If mtDNA depletion influences HAT function by limiting the overall cellular content of acetyl-CoA, as the data above suggest, then modulation of the mitochondrial pool of acetyl-CoA should correspondingly affect HAT activity. Thus, we next set out to test this by exposing rho+ and rho0 cells to pharmacological agents that modulate different entry points of the TCA cycle impacting the mitochondrial acetyl-CoA output, as previously described (Marino et al, 2014). In parallel, we evaluated HAT activity. We supplemented the medium of rho+ and rho0 cells with dimethyl-$\alpha$-ketoglutarate (DM-$\alpha$-KG), a cell permeable form of $\alpha$-KG that enters the mitochondria feeding the TCA cycle previously shown to increase acetyl-CoA (Marino et al, 2014). DM-$\alpha$-KG increased total cellular acetyl-CoA in rho0 but had no effects on rho+ (Fig 4A), and restored HAT activity in rho0 cells to levels similar to those in the rho+ counterparts (Fig 4B). Similarly, exposure of cells to dichloroacetate (DCA), an inhibitor of the mitochondrial pyruvate dehydrogenase kinases, also restored acetyl-CoA levels (Fig S3F) and HAT activity in rho0 but had no effects on rho+ cells (Fig 4C). Consistent with this result, direct activation of PDH by lipoic acid also increased HAT activity in rho0 cells (Fig S3G).

Conversely, diminishing the mitochondrial acetyl-CoA pool in rho+ cells by exposure to 1,2,3-benzenetricarboxylate (BTC) or perhexiline (PHX), inhibitors of the mitochondrial citrate carrier and carnitine transporter, respectively (Marino et al, 2014), reduced HAT activity (Fig 4D). BTC decreased total levels of acetyl-CoA in rho+ (Fig S5A). Interestingly, pharmacological inhibition of ACL with hydroxycitrate (HC), while diminishing acetyl-CoA and HAT activity in rho+, had no effects in rho0 cells (Fig 4E and F). These data confirm that the amount of cellular acetyl-CoA, whether decreased in the mitochondria or cytoplasm, influences HAT activity. They also support the notion that the levels of acetyl-CoA in rho0 are already limiting for ACL function.

**Figure 2. Genetic maintenance of histone acetylation prevents gene expression changes in the promoters of genes responding to acute mtDNA depletion.**
**(A)** Average genome-wide enrichment level of H3K9ac peaks (green lines) centered around the TSS of genes. Black lines show input DNA, which was used for normalization purposes. Red arrow indicates levels of average peak intensity at day 9. N = 2 per time point. **(B)** Log-fold enrichment levels of significantly changed H3K9ac peak tags at day 9 versus day 0 DN-POLG (blue) and NDI1/AOX (red) cells for genes falling on peaks within clusters A or B (as per Fig 1B). t test P < 0.01 is depicted by single asterisk (*) between statistical groups; equivalence testing between statistical groups was performed by the two one-sided tests (TOST) procedure. **(C)** Graphical representation of the H3K9ac peak densities at the TSS of four distinct DEGs identified in the DN-POLG cells (blues); the peaks corresponding to the same genomic loci in the NDI1/AOX counterparts are depicted in green. MTHFD1 (methylenetetrahydrofolate dehydrogenase 1), PSAT1 (phosphoserine aminotransferase 1), CBS (cystathionine beta synthase), and MLYCD (malonyl-CoA decarboxylase). Black bar indicates the gene and the vertical bars within depict the exons. **(D)** Genes within the genomic coordinates depicted in **(C)** were differentially expressed in the DN-POLG as previously reported (Lozoya et al, 2018); their expression levels as gauged by average fold-changes through microarrays in the NDI1/AOX-expressing cells at day 9 relative to day 0 is depicted. Statistical difference was evaluated with ANOVA. **(E)** Average fold-changes at day 9 relative to day 0 of genes associated with cholesterol biosynthesis (as per IPA analysis, Table S1) were calculated based on RNA-seq experiments performed in the DN-POLG cells (Lozoya et al, 2018). N = 3, statistical significance was gauged adjusting for multiple comparisons (FDR = 0.05). **(F)** Levels of lathosterol identified based on reanalysis of metabolomics data of DN-POLG and NDI1/AOX cells (Lozoya et al, 2018). Log2-fold change was calculated based on N = 4 for each cell line comparing levels at days 3, 6, and 9 to their respective day 0 counterparts; statistical significance was calculated based on two-way ANOVA and multiple comparison adjustments; error bars indicate ±SEM. **(G)** Heatmap of significantly represented canonical metabolic pathways per Ingenuity Pathway Analysis based on differentially enriched metabolites in DN-POLG and NDI1/AOX cells at day 9 of dox-inducible mtDNA depletion. Color intensity of heatmap blocks corresponds to strength of enrichment [−log(P) > 1.3]; red and green hues are representative of pathway enrichment based on predominance of up-regulated or down-regulated metabolites in the pathway, respectively.

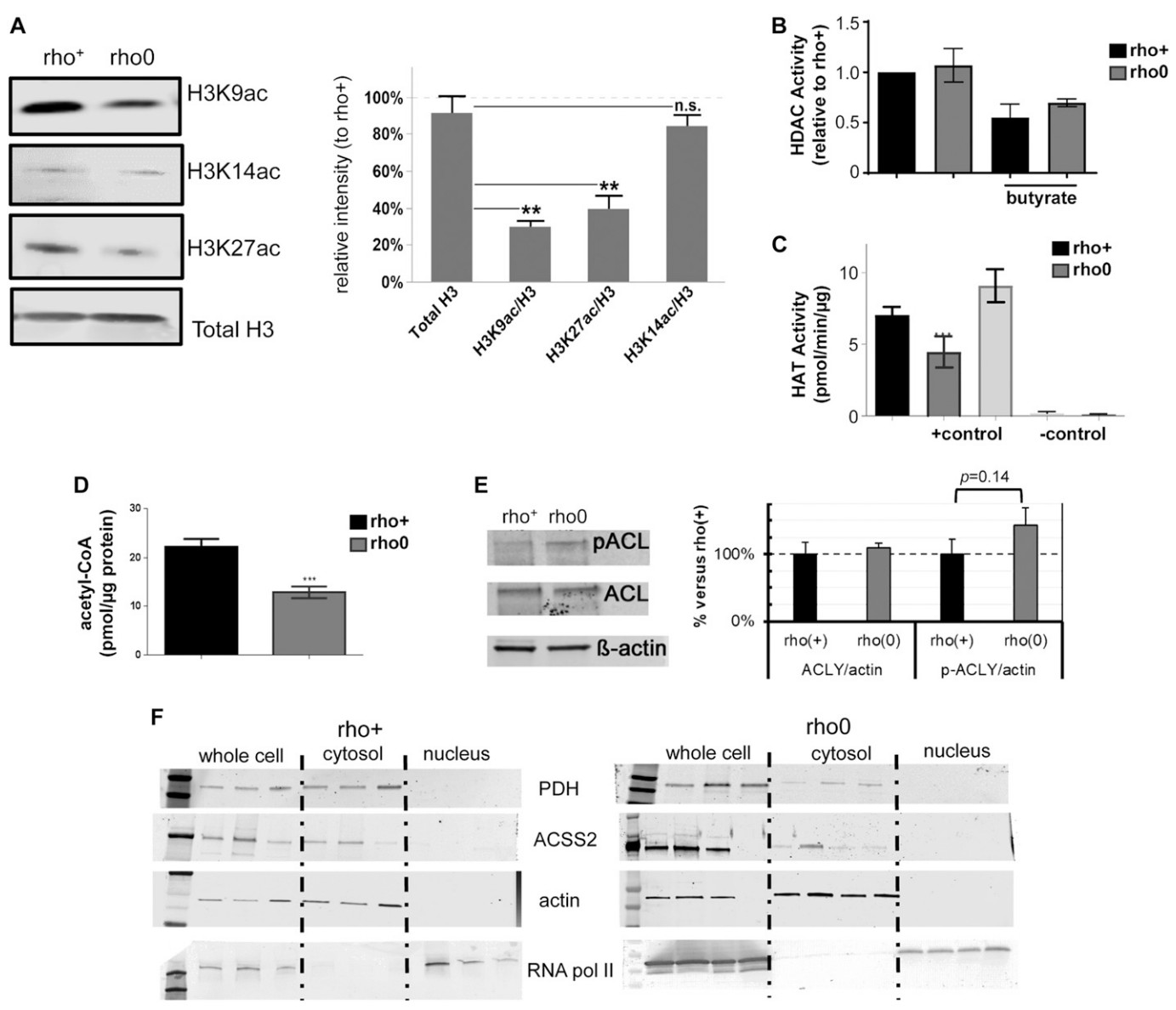

**Figure 3. Histones acetylation is decreased in H3K9 and H3K27 in chronically mtDNA-depleted cells because of decreased HAT activity.**
**(A)** Representative Western blots of different histone H3 acetylation marks in rho+ and rho0 cells; graph represents average data from N = 3 independent experiments. **(B)** HDAC activity was gauged based on a fluorometric assay using nuclear extracts of rho+ or rho0 cells (N = 6), including after 24-h incubation with 20 mM of butyrate to inhibit HDACs (N = 4). **(C)** HAT activity was estimated using a fluorescent assay and a standard curve (N = 5). Included were also a positive control provided by the assay and two negative controls that excluded acetyl-CoA or included assay buffer instead of sample. **(D)** Total levels of acetyl-CoA were estimated using deproteinized samples in a fluorescent-based assay and a standard acetyl-CoA curve. N = 3 biological replicates. Statistical analysis for HAT, HDAC, and acetyl-CoA measurements was performed using *t* test. **(E)** Representative Western blots probing phosphorylated and total ACL levels in three independent biological replicates of rho+ and rho0 cells. Graph depicts normalization of the data to actin for each antibody in each of the cell types. **(F)** Western blots probing levels of PDH and ACSS2 in whole cells and cytosolic or nuclear fractions of rho+ (left panels) and rho0 (right panels). β-actin was used as a marker of cytosolic and RNA Pol II of nuclear fractions, respectively. Western blots statistical differences were tested using ANOVA; error bars in all panels represent ±SEM.

## Reversal in locus-specific histone acetylation marks occurs in the promoters of genes whose expression is affected by DM-α-KG supplementation

In addition to modulating HAT activity, the above pharmacological interventions correspondingly altered histone acetylation (Fig S5B and C) providing the framework to interrogate cause–effect relationships between histone hypoacetylation caused by mtDNA depletion and gene expression regulation in the same cellular context. To address this question, we started by determining the genes differentially expressed between rho0 and rho+ cells by microarrays, which revealed about 3,300 DEGs (Table S3), and validation of random genes by quantitative real-time PCR is shown in Fig S6A. The metabolic pathways enriched by these 3,300 DEGs are shown in Fig 5A. Then, we supplemented the rho0 cells with DM-α-KG and performed microarray analysis of gene expression;

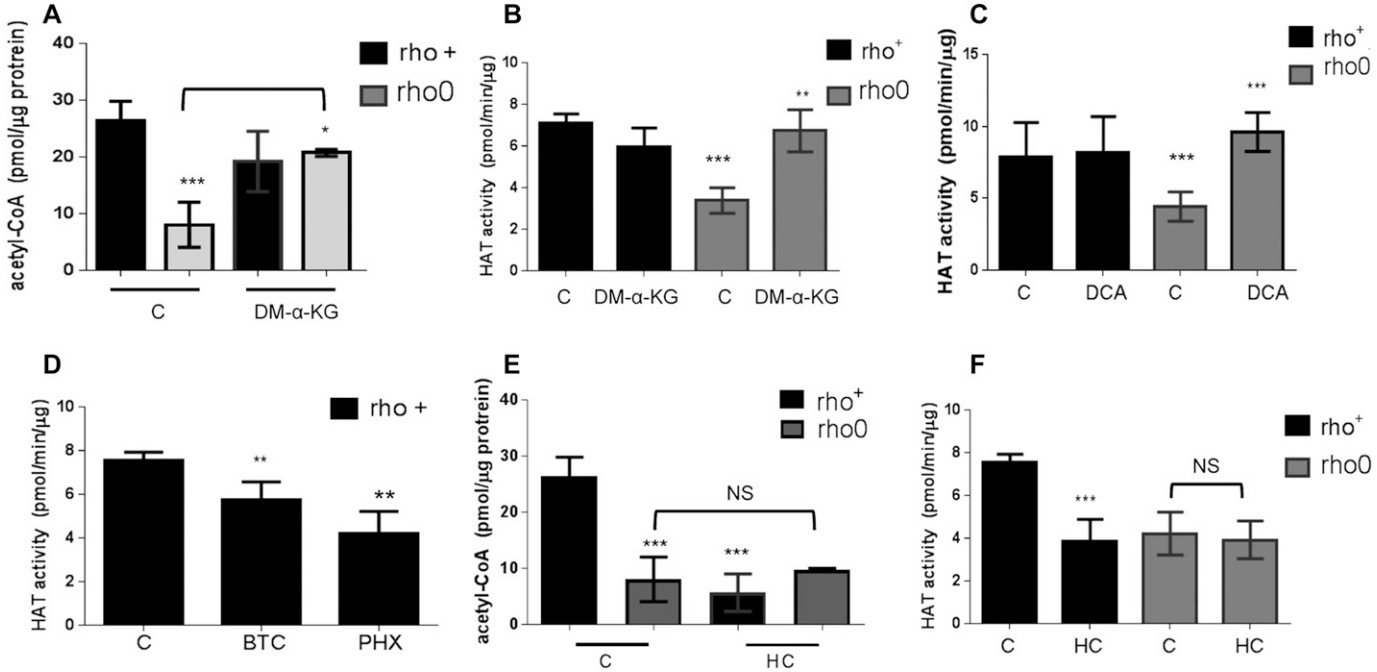

**Figure 4. Histone acetyltransferase activity can be modulated by altering the mitochondrial pool of acetyl-CoA.**
**(A)** Total levels of acetyl-CoA were estimated before or after exposure of rho+ or rho0 cells to DM-α-KG using deproteinized samples in a fluorescent-based assay and a standard acetyl-CoA curve. N = 3 biological replicates. **(B)** HAT activity was estimated in samples from (A) using an in vitro assay. **(C)** Rho+ and rho0 cells were exposed to the mitochondrial pyruvate kinase inhibitor DCA for 4 h before HAT activity estimation using a fluorescent-based assay; N = 3. **(D)** Rho+ cells were treated with the mitochondrial citrate carrier inhibitor BTC or an inhibitor of the mitochondrial carnitine transporter, perhexiline (PHX), and HAT assays performed as in (C). **(D)** Rho+ or rho0 cells were exposed to hydroxycitrate (HC), an inhibitor of the cytosolic ACL, for 4 h. Total acetyl-CoA levels were then estimated using deproteinized samples in a fluorescent-based assay and a standard acetyl-CoA curve. N = 3 biological replicates. **(F)** HAT activity was gauged in the samples from **(E)**. Statistical analysis for HAT and acetyl-CoA measurements was performed using *t* test; error bars in all panels represent ±SD.

DM-α-KG was chosen because it needs to be metabolized in the mitochondria to generate acetyl-CoA (Marino et al, 2014). Also, under our experimental conditions, it increased acetyl-CoA and HAT activity in rho0 cells but had no effects on rho+ cells (Fig 4A and B). We found that 596 of the 3,295 DEGs in rho0 cells had their expression changed by the DM-α-KG treatment (Fig 5B and Table S4). DM-α-KG partially or fully rescued the directionality of the change for ~70% of the 596 affected genes (Fig 5B and Table S4), which was noteworthy given the chronic state of transcriptome remodeling in the rho0 cells.

Pathway enrichment using Ingenuity Pathway Analysis (IPA) revealed that the 596 DM-α-KG–sensitive genes in rho0 cells were broadly involved in metabolic or signaling pathways; a schematic representation of a few enriched pathways is shown in Fig 5C and the full list can be found in Fig S6B and C. The transcriptional response of some genes involved in cellular metabolism was not surprising, given that many can be affected by changes in acetyl-CoA levels. For instance, SAT1 is an enzyme involved in the catabolism of spermidine and spermine, a reaction that consumes cytosolic acetyl-CoA. SAT1 is known to be regulated transcriptionally and to be dependent on the levels of putrescine, the catabolic byproduct of spermidine/spermine (Pegg, 2008). SAT1 was down-regulated in rho0 relative to rho+ cells, consistent with decreased acetyl-CoA availability, but less so upon DM-α-KG exposure (Table S4). Conversely, the sensitivity of genes associated with the immune response and inflammation, which were mostly up-regulated in the

rho0 cells after DM-α-KG, was unexpected given that there is no reported direct link between these pathways and acetyl-CoA levels or DM-α-KG metabolism. However, connections between these phenotypes and mitochondrial dysfunction do exist.

Prediction of upstream regulators of these 596 DM-α-KG–sensitive genes using ChIP-seq data from the ENCODE consortium through Enrichr (Chen et al, 2013; Kuleshov et al, 2016) identified the histone acetyltransferase EP300 as the only hit with a significant adjusted *P*-value; no other epigenetic effectors that could be affected by α-KG, such as the demethylases of DNA or histones, were identified (Fig S6D). This is consistent with the hypothesis that histone acetylation changes contribute to regulate gene expression in the context of mitochondrial dysfunction. Remarkably, about 60% (352/596) of the genes affected by DM-α–KG were found in EP300 ENCODE ChIP-seq experiments (Table S4). α-KG can also affect the hydroxylases that stabilize HIF, and although HIF1α was not enriched using the ENCODE data (Fig S6D), its protein levels were slightly increased by this treatment (Fig S6E). Detailed analysis revealed that only 15 of the 596 DM-α-KG–sensitive genes are HIF1α targets (Table S4), leading us to conclude that its contribution to the differential expression of genes upon DM-α-KG exposure was minor.

To directly test whether changes in promoter histone acetylation influenced the expression of those 596 genes, we performed ChIP-seq in rho0 cells before and after supplementation with DM-α-KG. We used antibodies against H3K9ac and H3K27ac as both of these

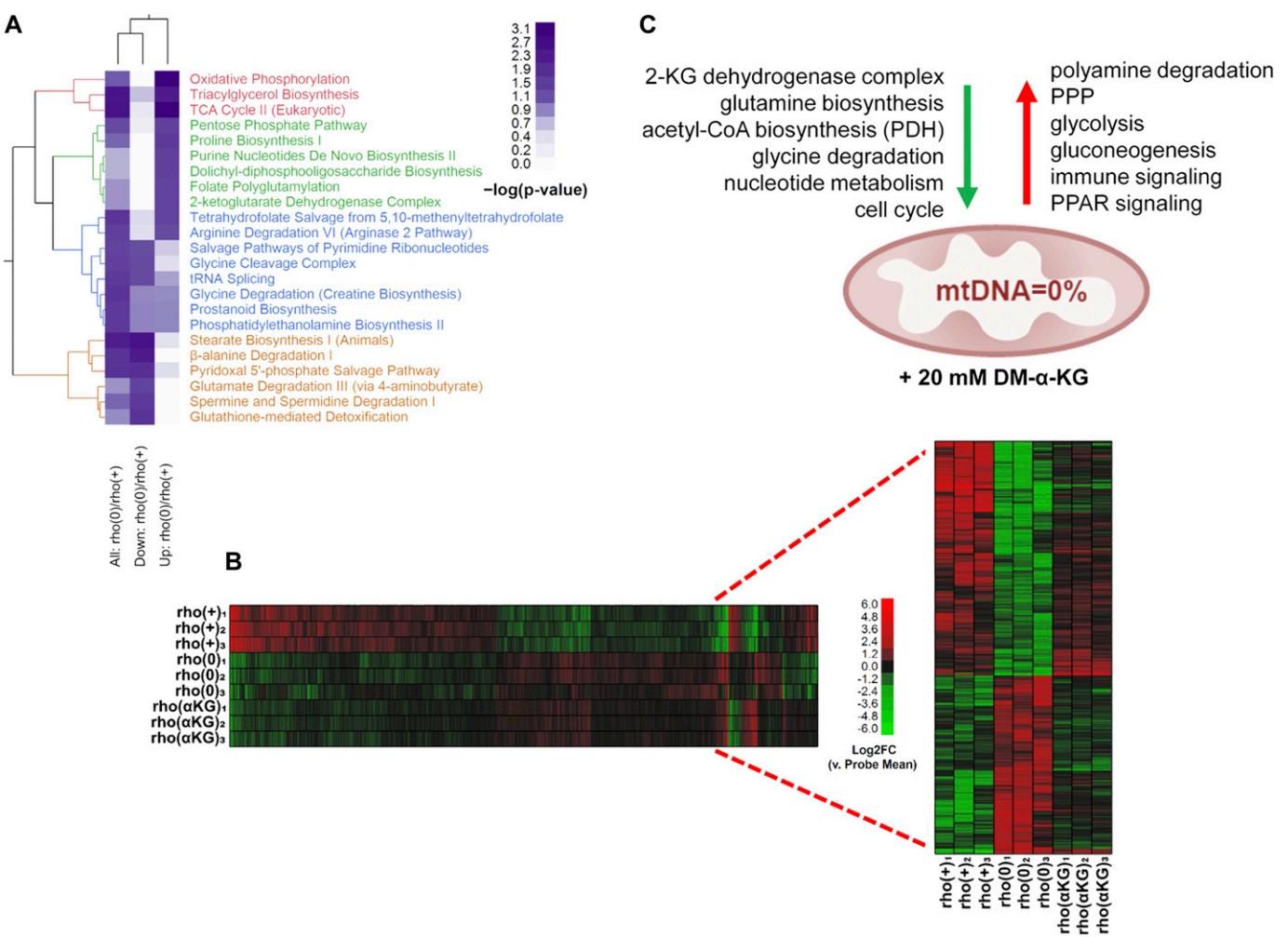

**Figure 5. Treatment with DM-α-KG rescues gene expression in the context of chronic mtDNA depletion.**
**(A)** Heatmap of significantly represented canonical metabolic pathways per Ingenuity Pathway Analysis based on DEGs when comparing rho+ and rho0. Color intensity of heatmap blocks corresponds to strength of enrichment [−log($P$) > 1.3]. **(B)** Microarrays were performed using RNA from rho+, rho0, or rho0 pretreated with DM-α-KG; N = 3 per sample. Inset show data from the 596 genes that were impacted by the treatment. **(C)** Schematic representation of the main transcriptional responses reversed in the rho0 cells by DM-α-KG exposure; green and red arrows indicate down-regulated and up-regulated signaling pathways, respectively, after treatment.

marks are associated with promoters; H3K27ac marks also map to enhancer regions (Zhou et al, 2011). For this analysis, we followed the same rigorous protocol as used for the DN-POLG analysis, first determining the number of peaks identified relative to input DNA. We found a total of 25,340 H3K9ac and 40,885 H3K27ac peaks in rho0 cells and lower number after treatment with DM-α-KG: 24,117 for H3K9ac and 30,548 for H3K27ac (Fig 6A). Similar findings were obtained when evaluating the area under the curve (AUC) as a metric for peak abundance (Fig S7A). No effects of DM-α-KG over input DNA were identified, irrespective of the metric used (Fig S7A and B), which rules out the possibility that the decrease in peak numbers simply reflected changes in normalization parameters. Although the genome-wide changes in the H3K9ac peak numbers were most prominent in gene bodies, those for H3K27ac were also found in intergenic regions. The decreases in total number of peaks by the DM-α-KG was unexpected but given that H3K9 and

H3K27 residues can also bear other posttranslational modifications, these results may reflect a change in the type of lysine modification associated with some genomic locations. Irrespective of this, the number of peaks in the promoter regions of all genes was similar (Fig 6A).

We started by evaluating H3K9ac or H3K27ac peak status before and after DM-α-KG exposure in all 596 genes, independent of whether they were up or down-regulated (Table S4). The behavior of H3K9ac and H3K27ac peaks was similar when evaluating average enrichment across the promoter coordinates of the 596 genes (Fig S7B), although enrichment of H3K27ac was more predictive of changes in gene expression (Fig S7B). The mean enrichment levels of promoter peaks for the subset of 291 genes up-regulated in rho0 cells, but rescued by DM-α-KG, were significantly decreased on their TSSs; similar effects were observed for the 101 genes that were further down-regulated in rho0 cells after the treatment with DM-

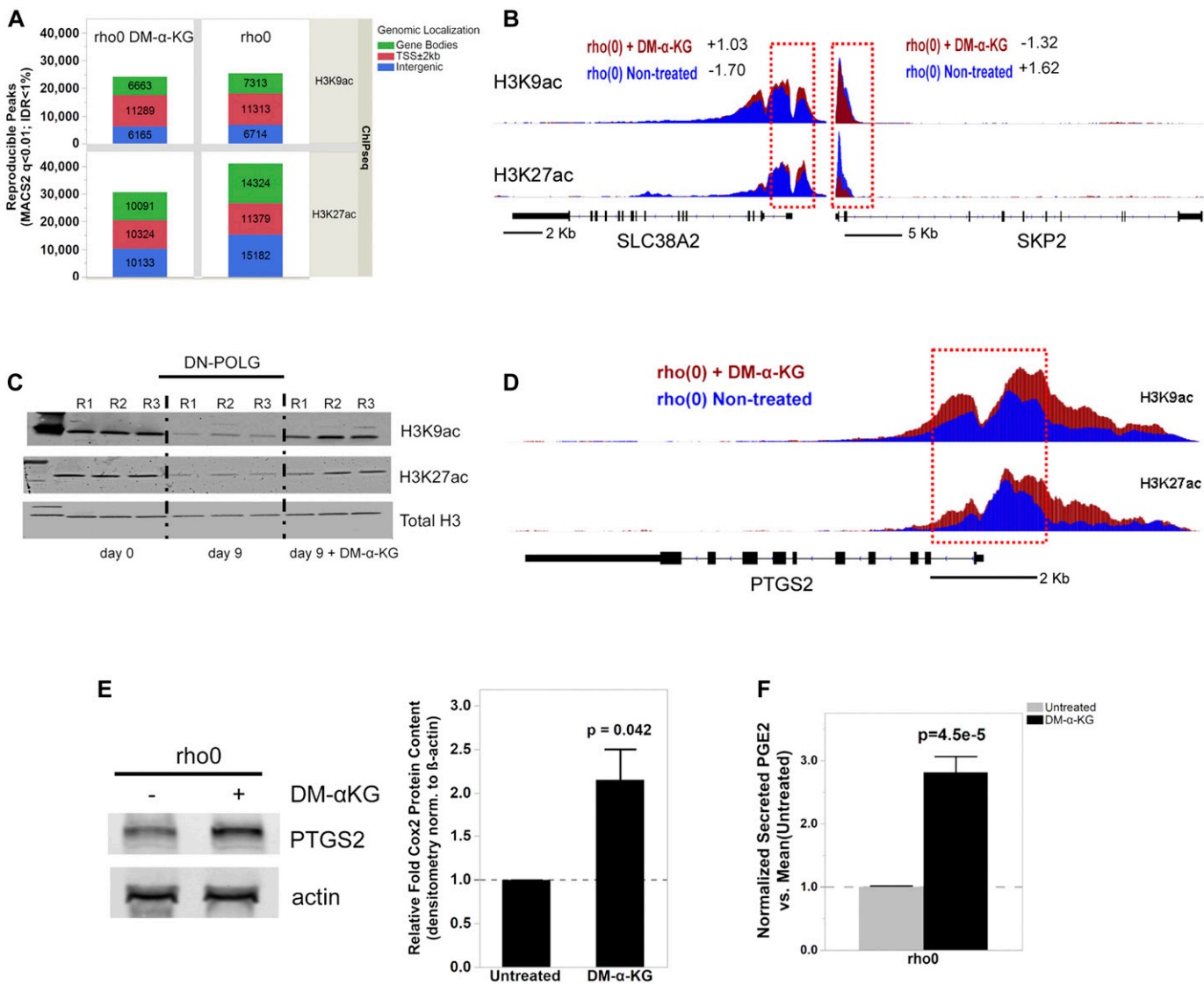

**Figure 6. Exposure to DM-α-KG reverses locus-specific H3Kac marks in 143B rho0 cells on the genes affected by the same treatment.**
**(A)** Stacked bar plots of genomic localization for H3K9ac (top) and H3K27ac (top) detected reproducible peaks (q < 0.01, IDR < 1%) by ChIP-seq with respect to gene coordinates in 143B rho0 cells with or without 4 h supplementation with 20 mM DM-α-KG in culture. **(B)** Graphical representation of the H3K9ac and H3K27ac RPM values in two genes differentially expressed in rho0 cells (relative to rho+) with (red) and without (blue) DM-α-KG supplementation; black bar indicates the gene and the vertical bars within depict the exons. Numbers reflect the fold-change relative to rho+ in non-treated or DM-α-KG-exposed rho0 cells. Tracks for SLC38A2 (solute carrier family 38 member 2) and SKP2 (S-phase kinase-associated protein 2) are representative. **(C)** DN-POLG cells were treated with doxycycline to induce mtDNA depletion, and at day 9, the cells were exposed for 4 h to 20 mM DM-α-KG. Histones were extracted and acetylation levels of H3K9 and H3K27 were probed using antibodies; data show immunoblots of three independent cultures (R1–R3). Total histone H3 was used as loading control. **(D)** Same as (B) but for the PTGS2 gene. **(E)** Representative Western blots of PTGS2 in rho0 cells before and after 4-h exposure to DM-α-KG; graph shows mean of three independent biological replicates. **(F)** Levels of PGE2 were estimated using ELISA in the supernatant of cells used for (E) N = 3. Data were normalized to total protein content; ANOVA was used to gauge statistical differences; error bars represent ±SEM (E, F).

α-KG (Fig S7B, left panels). These results were consistent with the fact that histone hypoacetylation is associated with decreased chromatin accessibility to transcription factors (Gorisch et al, 2005). Conversely, H3K27ac peak enrichment increased for the 86 genes up-regulated in rho0 cells (Fig S7B, lower right panels). In addition to this average peak intensity, we further analyzed H3K9ac and H3K27ac peaks on specific genes, with representative examples depicted in Fig 6B. Taken together, these data show that the 596 genes whose expression in rho0 cells that were sensitive to DM-α-KG showed

concomitant reversal of their promoter H3K9ac and/or H3K27ac status. Importantly, treatment of DN-POLG cells depleted of their mtDNA at day 9 with DM-α-KG also increased histone acetylation (Fig 6C). Collectively, the results obtained with the DM-α-KG experiments together with the ChIP-seq and gene expression analysis on the DN-POLG cells strongly support a model where the histone acetylation changes driven by mtDNA depletion regulate gene expression.

We next asked if the locus-specific H3Kac and gene expression changes driven by DM-α-KG supplementation would lead to

measurable functional outcomes. To exemplify this relationship, we chose prostaglandin G/H synthase 2, also known as cyclooxygenase-2 (PTGS2 or Cox2), because this gene was decreased in rho0 versus rho+ cells, and its expression was completely recovered in rho0 relative to rho+ cells after DM-$\alpha$-KG treatment (Table S4). PTGS2 converts arachidonate to prostaglandin E2 (PGE2), which can be measured in the tissue culture supernatant. Concomitant to the increased gene expression in rho0 cells after DM-$\alpha$-KG treatment was the significant increase in promoter abundance of the H3K9ac and H3K27ac marks (Fig 6D). Changes in PTGS2 mRNA in rho0 cells led to a parallel increase in protein (Figs 6E and S7C), which resulted in significant increases in the levels of secreted PGE2 after exposure to DM-$\alpha$-KG (Fig 6F). No significant changes in protein or PGE2 levels were observed in rho+ exposed to DM-$\alpha$-KG (Fig S7D and E). Therefore, these data strongly link mitochondria-driven changes in histone acetylation and gene expression to functional outcomes.

From an epigenetic perspective, DM-$\alpha$-KG could affect the methylation status of the epigenome since $\alpha$-KG is a cofactor of enzymes that drive demethylation reactions, including histone demethylases and the Ten Eleven Translocation (TET) enzymes (Hitchler & Domann, 2012). However, no decreases in the levels of H3Kme3 marks were found in the rho0 cells treated with DM-$\alpha$-KG (Fig S8A). Similar results were obtained when evaluating the DN-POLG cells (Fig S8B). We used dot blots to estimate the levels of 5hmeC, the byproduct of TET activity, on genomic DNA of rho0 before and after exposure to DM-$\alpha$-KG but did not find changes in its levels (Fig S8C). We recently performed DNA methylation arrays at a single nucleotide resolution in these rho0 cells and showed that 621 DEGs were differentially methylated compared with the rho+ cells (Lozoya et al, 2018). Thus, we reasoned that if DM-$\alpha$-KG affected DNA demethylation in a way that would reverse gene expression under our experimental conditions, then the loci associated with those genes should start out by being hypermethylated in rho0 cells. Of the 596 DM-$\alpha$-KG–sensitive genes, 101 were differentially methylated in rho0 relative to rho+ cells (Table S5), but only 39 of these started out as being hypermethylated. Thus, if differential DNA methylation also played a role in the effects associated with DM-$\alpha$-KG treatment, only a small portion of genes that showed reversed expression (6.5%) may have had their transcription influenced by this epigenetic modification.

## Discussion

Mitochondrial function is key to organismal health, and it is now accepted that mitochondria affect cellular physiology through mechanisms beyond bioenergetics and reactive oxygen species. However, the means through which modulation of mitochondrial metabolism can effectively change biological outcomes is still under investigation. Because various mitochondrial TCA cycle metabolites are either substrates or cofactors of enzymes that impact the epigenome, presumably mitochondrial dysfunction or mitochondrial metabolic rewiring can impact epigenetic regulation of gene expression. Although we previously reported on the link between mitochondrial dysfunction and nuclear DNA hypermethylation

(Lozoya et al, 2018) no study so far has demonstrated the requirement of mitochondrial function for long-term maintenance of chromatin acetylation. Moreover, to our knowledge, there is no previous available evidence showing that changes in histone acetylation can be influenced by the modulation of the mitochondrial output of acetyl-CoA in a way that can affect HAT function and the expression of genes.

Here we report that under mtDNA depletion, steady state levels of acetyl-CoA and histone acetylation were decreased, which was observed both when mtDNA was progressively lost (DN-POLG cells) or chronically depleted (143B rho0). We also show that these effects were associated with the mitochondrial output of acetyl-CoA, which in turn influenced HAT activity in a reversible way, overall contributing to the regulation of gene expression in the nucleus. Several of these findings were unexpected. First, the maintenance of H3K9 and H3K27 primarily in the hypoacetylated state under chronic mitochondrial dysfunction suggest that unlike the response to loss of ACL function (Zhao et al, 2016), loss of acetyl-CoA associated with mtDNA depletion is not compensated for. At least in terms of gene expression, we identified no obvious means to increase acetyl-CoA production or spare acetyl-CoA utilization, including through acetate or lipid metabolism or enhanced ACL transcription (Table S3). We also did not identify changes in protein levels or subcellular localization of PDH or ACSS2 in rho0 cells. Alternatively, some level of compensation may exist, but perhaps maintenance of histone acetylation may be secondary to other non-histone proteins, particularly in the context of mitochondrial dysfunction. This is a compelling possibility considering that acetylation is the second most abundant posttranslational modification of proteins in cells, with ~90% of metabolic enzymes estimated to be regulated by it (Drazic et al, 2016), including 20% of the mitochondrial proteome (Kim et al, 2006). Along those lines, others have proposed a role for histones as reservoir of acetate that can be used for metabolic processes, such as lipogenesis, under specific conditions (Fan et al, 2015). It is possible that maintenance of cellular or even mitochondrial function itself relies on a steady state level of acetylation reactions above a minimal functional threshold.

Second, the findings that HAT activity was influenced by the mitochondrial output of acetyl-CoA have not been previously reported. HATs can be regulated by substrate/cofactor availability, protein–protein interactions and posttranslational modifications, including acetylation (Legube & Trouche, 2003). Although it is known that some HATs have kinetic properties that would allow them to respond to fluctuations in the levels of acetyl-CoA (Fan et al, 2015, Kuo & Andrews, 2013), our in vitro HAT assays were carried out in the presence of excess acetyl-CoA, making it unlikely that substrate availability was limiting. Nevertheless, in the cellular context, this could still be formally possible. It is likely that the decreased levels of acetyl-CoA under our experimental conditions may have changed the posttranslational state of some HATs or other proteins with which they interact within the context of chromatin structure. Irrespective of the means, the effects of pharmacological manipulation of the mitochondrial pool of acetyl-CoA, both in the rho0 and in the rho+ cells, on HAT function unveiled a novel mechanistic link between mitochondrial metabolism and histone acetylation. It is interesting that we found here and in our

previous work (Martinez-Reyes et al, 2016) that changes in abundance of lysine acetylation only occurred in some residues on H3, H2B, or H4 when mtDNA was depleted. This is contrary to the work that showed glucose deprivation led to global histone deacetylation in H3, H2B, and H4 (Cluntun et al, 2015). It is not clear why an overall cellular decrease in acetyl-CoA provided by mitochondrial dysfunction would affect some lysine residues and not all. Interestingly, fluctuations in acetyl-CoA were shown to affect both global levels of histone acetylation and the pattern of acetylated residues by altering lysine acetylation preference by some HATs (Henry et al, 2013). It is possible that cellular metabolism is impacted in fundamentally different ways when mitochondria are dysfunctional compared with glucose withdrawal, overall affecting HATs in distinct ways.

Third, the genetic or pharmacological modulations of the TCA cycle flux in the HEK293 DN-POLG or in the 143B rho0 cells provide the most compelling data to connect mitochondrial TCA cycle function with changes in histone acetylation, gene expression and physiological outcomes with cause-effect relationships. This is supported by recent work from another group that showed that mitochondrial metabolism of acetate, pyruvate, and fatty acids influence H3K27ac levels and promote cell adhesion (Lee et al, 2018). Under our experimental conditions, the fact that expression of 20% of the DEGs was rescued by a short 4-h exposure to DM-$\alpha$-KG was remarkable, given that these cells were in a state of chronic mitochondrial dysfunction with a well-established pattern of gene expression. The modest changes observed in the levels of histone acetylation after treatment with DM-$\alpha$-KG may explain why only a fraction of the genes were affected. It is possible that longer treatments with DM-$\alpha$-KG could have more pronounced effects. Alternatively, perhaps only some genes would be responsive to such intervention. Increasing cellular levels of acetyl-CoA through modulation of TCA flux could also impact generalized protein acetylation, including that of transcription factors. Likewise, the overall metabolic rescue provided by our interventions could play a direct role in turning off the signals for differential gene transcription. Considering that we found that both metabolic and other categories of genes were differentially expressed, we do not favor this possibility.

About 70% of the DEGs had an altered H3K9ac mark in the 9-d course of mtDNA depletion in the DN-POLG system; changes in this epigenetic mark were apparent already at day 3. Because the genomic changes at day 3 precede signs of measurable mitochondrial dysfunction (Martinez-Reyes et al, 2016), these data suggest that histone acetylation is highly sensitive to alterations in mitochondrial metabolism and may eventually prove to be a biomarker of mitochondrial dysfunction. In addition, because broad range transcriptional variations were not identified at day 3, these data suggest that substantial remodeling of the histone acetylation landscape may be required for changes to the transcriptional program. Notably, not all DEGs had alterations in their promoter H3K9ac levels. Likewise, many genes that were transcribed but were not differentially expressed did show changes in H3K9ac abundance, which was also the case when we monitored modulation of DNA methylation (Lozoya et al, 2018). Although we found that a gene was statistically more likely to be differentially expressed if it also had a change in one of its associated epigenetic marks, what these

data highlight is that the epigenetic alterations that we monitored are not by themselves sufficient to drive the differential expression of genes.

Genome-wide effects of metabolite-driven modulation of epigenetic marks paralleled by a specific transcriptional output in only a fraction of the affected loci is not uncommon, raising fundamental questions about how specificity can be achieved when the entire landscape is reprogrammed (Schvartzman et al, 2018). Although addressing this question is still challenging, we propose that the large-scale epigenetic effects provided by mitochondrial dysfunction that we report here and in our previous work (Lozoya et al, 2018) may alter 3D chromatin configuration and the spatial organization of genes in a way that facilitates the recruitment of specific transcription factors, epigenetic writers, readers, and erasers. Additional studies involving chromosome conformation capture techniques such as Hi-C, ChIP-loop or ChIA-pet can help address this possibility.

In summary, our studies uncovered a significant and yet unappreciated means through which mitochondrial function can impact the cell. Additional experiments to try to identify which HATs and dissect the exact means through which they are affected by the mitochondrial output of acetyl-CoA will be fundamental in defining more in depth the mechanistic link between mitochondrial metabolism and histone acetylation. However, our findings provide a new foundation to address the phenotypic heterogeneity and tissue-specific pathology associated with mitochondrial dysfunction, including that which is caused by mitochondrial genetic diseases. These results may not only be relevant to the pathophysiology of mtDNA depletion syndromes but also to understanding the biological mechanisms of environmental agents that lead to mtDNA loss, such as nucleoside reverse transcriptase inhibitors that are used to treat HIV infections. Our results may also be pertinent to the effects of many antibiotics that, by inhibiting mitochondrial protein synthesis, lead to functional effects comparable with mtDNA loss. Our findings that pharmacological promotion of the TCA cycle by supplementation with DM-$\alpha$-KG can rescue histone acetylation and the differential expression of many genes associated with chronic mitochondrial dysfunction could provide the underlying mechanism for the improvement in muscle strength in a mouse model of mitochondrial myopathy by a 5-mo supplementation with 5 mM $\alpha$-KG recently reported (Chen et al, 2018). Assuming that such a mechanism is broadly applicable *in vivo*, this could eventually prove to be a novel approach for therapeutic purposes. Although the concentration of DM-$\alpha$-KG used in our and other studies (Chen et al, 2018) is well beyond physiological levels, these proof-of-concept experiments open challenges and opportunities regarding metabolic modulation of mitochondrial function to impact health and disease.

# Materials and Methods

## Cells, cell cultures, and experimental conditions

The DN-POLG and NDI1/AOX cells were described recently and maintained according to the previous published work (Martinez-

Reyes et al, 2016). The osteosarcoma cell line 143B and the rho0 derivative were graciously obtained from Dr. Eric Schon at Columbia University and were routinely grown in DMEM high glucose (4.5 g/l) supplemented with 10 mM pyruvate, 50 µg/ml of uridine, 10% FBS, and 1% penicillin/streptomycin under 37°C and 5% $CO_2$. The freshly mtDNA depleted cells used for Fig S3A–C were obtained by exposing the 143B control cells to EtBr (50 ng/ml) for 2 wk. TFAM MEFs were a kind gift from Dr. Gerald Shadel (Yale School of Medicine). TFAM MEFs media contained high glucose and pyruvate but did not contain uridine. All experiments were carried out with confluent cell cultures. Acetyl-CoA manipulations using hydroxycitrate, PHX, DCA, and BTC were performed as previously described (Marino et al, 2014) with DM-$\alpha$-KG concentrations between 5 and 20 mM; the pH of final drug concentrations was adjusted to 7.0 using NaOH.

## ATP, acetyl-CoA, NAD$^+$/NADH ratio, and $H_2O_2$ measurements

Cells were pelleted and resuspended in 3.5% of perchloric acid. Lysates were then sonicated on ice for 30 s, flash-frozen in liquid nitrogen, thawed on ice, and then span at 5000 $g$ at 4°C for 10 min. Supernatants were then collected, neutralized to pH ~7.2 with 1 M KOH, and 50 µl used for the ATP and acetyl-CoA assays (BioVision) following the manufacturer's instructions. Levels of these metabolites were estimated based on standard curves; data were normalized to protein content obtained from parallel cell cultures. The NAD$^+$/NADH ratio was determined using an equal number of cells (10,000) per cell type with a kit from Promega. Amplex Red (Invitrogen) was used to detect the levels of $H_2O_2$ released in medium as previously described (Santos et al, 2003).

## Histone preparations and Western blots

Histones were purified from nuclear lysates using trichloroacetic acid and acetone as described by others (Shechter et al, 2007). Relative amounts of modified histones were assayed by SDS–PAGE immunoblotting in multiple independent biological replicates (N ≥ 3 per cell derivative). Total histone protein content per sample was estimated from parallel immunoblots to detect total H3 signal using a pan-specific primary antibody as loading control. All histone antibodies were obtained from Active Motif (Millipore) or LiCor (secondary antibodies). For PTGS2, antibodies were purchased from Santa Cruz Biotechnology; primary antibodies raised against total or phosphorylated ACL protein, PDH, and ACSS2 were obtained from Cell Signaling. HIF1a antibodies were purchased from Abcam, and GCN5, RNA, pol II and EP300 were obtained from Active Motif.

## HAT and HDAC activities

Cells were lysed and the nuclear fraction enriched using differential centrifugation. HAT activity was gauged using 1 µg of nuclear lysates and a fluorometric assay (BioVision) following the manufacturer's instructions. HDAC activity was assayed using 1 µg of nuclear lysates and a fluorometric kit available from Enzo Life Sciences following the manufacturer's instructions. Data were normalized to protein content.

## PGE2 content

Levels of secreted PGE2 were estimated using an ELISA kit following the manufacturer's instructions (Cayman Chemical). Data reflect three independent biological replicates and were normalized to total protein content.

## Gene expression experiments by microarray technology and data analyses

For microarrays analysis of gene expression, the Affymetrix Human Genome U133 Plus 2.0 GeneChip arrays were used. Samples were prepared as per manufacturer's instructions using total RNA. Arrays were scanned in an Affymetrix Scanner 3000 and data were obtained using GeneChip Command Console and Expression Console Software (AGCC; Version 3.2 and Expression Console; Version 1.2) using the MAS5 algorithm to generate CHP-extension files. ANOVA was used to identify statistical differences between means of groups at $\alpha$ < 0.05 level among HG-U133 Plus 2.0 probe sets unambiguously mapped to UCSC known gene transcripts. Gene expression patterns were determined empirically from log$_2$-transformed expression fold-changes by unsupervised hierarchical clustering (Ward's distance metric) using JMP software (Version 11).

## ChIP-seq and data processing

Chromatin immunoprecipitations were performed with modifications as recommended elsewhere (Carey et al, 2009; Fujita & Wade, 2004). Briefly, cells grown in adherent monolayers (20–30 million per individual replicate, N = 2) for each of the 143B rho+ and rho0 derivatives, or the DN-POLG and NDI1/AOX derivatives, were cross-linked by addition of paraformaldehyde at a 1% (vol/vol) final concentration directly to culture media followed by 8-min incubation at room temperature; chemical cross-linking was quenched by further supplementation with glycine at 125 mM. Cross-linked cells were scraped off the plates, washed in PBS, pelleted by centrifugation at 4,000 $g$ and 4°C for 10 min, homogenized in 2 ml of lysis buffer containing protease inhibitors (Halt; Thermo Fisher Scientific), and sheared with a temperature-controlled BioRuptor instrument with high-power settings at 4°C for 20–25 1-min pulses (50%-duty cycle) of ultrasonic shearing. Sheared cell homogenates from two biological replicates containing 1–10 µg DNA each were used to extract control (10% input DNA) and ChIP DNA templates with antibodies against H3K9ac or H3K27ac (Active Motif). Each template was ligated and amplified into sequencing libraries using different single-indexed adapters (TruSeq RNA v2, Set A; Illumina). Each individual library was PCR-amplified in two technical replicates for no more than 10 cycles (screened per sample by RT-PCR as the number of cycles to reach the inflection point in log-scale amplification curves); afterwards, duplicate PCR reaction volumes per sample were collected for DNA purification with size selection by double-sided 0.6×–0.8× SPRI (expected fragment size: 250–450 bp inclusive of sequencing adapters) using AMPureXP magnetic beads (Beckman-Coulter). On surveying library quality control, which was performed in four-plexity runs of 20–30 million 2 × 35-nt paired-end reads in an MiSeq system (Illumina), the samples were sequenced in 8-plexity runs using a NextSeq 500 system with high-output flow

cells, (Illumina) following the manufacturer's protocols. Adapter 3' sequences were removed from raw ChIP-seq reads after filter for quality phred scores > 20, followed by 5' trimming of bases 1–10 of each read. The resulting paired-end 2 × 25-nt reads were aligned to the hg19 human reference genome (Genome Reference Consortium GRCh37 from February 2009) (Kent et al, 2002) with Bowtie 2—sensitive local settings and 1,000-bp maximum fragment length (–D 15 –R 2 –N 0 –L 20 –i S,1,0.75 –X 1000) (Langmead & Salzberg, 2012). Within-sample consistency between replicative sequencing runs from individual ChIP templates was confirmed based on Pearson pairwise correlation scores of log-transformed uniquely mapped and de-duplicated aggregate RPKM values across ~18,000 non-overlapping 10-Kb genomic regions centered around gene TSS. To assess levels of between-replicate concordance of H3K9ac or H3K27ac peak enrichment in each experimental group, we implemented ChiLin (Qin et al, 2016) with paired-end mapping (Bowtie 2) (Langmead & Salzberg, 2012) and narrow peak-calling modes (MACS2, false discovery rate [FDR] q < 0.01 v. input DNA) (Zhang et al, 2008). Peak calling was followed by estimation of reproducible peak numbers via bivariate ranking of narrow-peak enrichment significance at the 1% irreproducible discovery rate (IDR) level (Li et al, 2011); when detected across samples, overlapping peaks were merged into consensus genomic tags of differential H39ac or H3K27ac occupancy. Lists containing all consensus genomic loci detected across experimental groups, with differential H3K9ac or H3K27ac peak enrichment, and passing the IDR < 1% threshold were assembled into a conglomerate set of reference genomic tags for further data analysis using SeqMonk, version 37.1 (Andrews, S. SeqMonk, 2007: http://www.bioinformatics. babraham.ac.uk). For quantitative comparisons, tag densities of H3K9ac peaks detected at each time point and in each independent biological replicate were normalized against tag densities in matching input DNA control libraries sequenced in parallel (Fig 1A black lines; see the Materials and Methods section for details). In all cases, we confirmed consistency across individual sample libraries for genome-wide read count distributions before further study (Nakato & Shirahige, 2017). Libraries included in the analysis were also scored for compliance with proposed quality metrics per the ENCODE Project; these were implemented via the ChiLin quality control pipeline (Qin et al, 2016), and included fraction of reads in peaks, cross-correlation profiles between replicates for signal to noise ratio measurements, and peak coincidence cross-replicates among others. Reproducible peaks used in the analysis were extracted via bivariate ranking of narrow-peak enrichment significance with a 1% IDR cutoff (Li et al, 2011). To gage the correspondence of H3K9ac and H3K27ac peak enrichment (ChIPseq data) with gene expression sensitivity to DM-$\alpha$-KG supplementation (Affymetrix Human Genome U133 Plus 2.0 GeneChip array data), the AUC (AUC = reads per million (RPM)-adjusted ChIP minus RPM-adjusted background, length adjusted; as on Mews et al [2017]) of any ChIP-seq tags detected in any of the biological groups were calculated, assigned to the nearest protein-coding human genome organization nomenclature committee-recognized expressed gene within a distance of 2 kb relative to annotated TSS coordinates, and grouped by the differential expression status and DM-$\alpha$-KG sensitivity of their nearest gene

assignment. ChIPseq tags beyond 2 kb from TSS coordinates of expressed genes were not considered for the analysis.

### Lysine acetyltransferase gene mutation analysis

To find SNPs in rho0, we used DNA reads sequenced with a MiSeq system (Illumina) and aligned to the hg19 human reference genome (Genome Reference Consortium GRCh37 from February 2009) (Kent et al, 2002). The human genome sequences were downloaded from ENSEMBL ftp site (Herrero et al, 2016). Alignments were performed using Bowtie 2 (Langmead & Salzberg, 2012). We then parsed alignment results from the coordinates of the 17 genes from lysine acetyltransferase family, each security account manager (SAM/ binary storage of SAM (BAM) extension file into pileup data using SAMtools (Amoozegar et al, 2009). The coordinates of each gene on hg19 were taken from NCBI's Gene database (Coordinators, 2016). An SNP was called if more than 80% of reads containing a change from original nucleotide aligned at call position. A Perl script was developed to perform the parsing CIGAR field of the mpileup file into Variant Call Format (VCF), to extract all SNPs and insertions/deletions (INDELs). No INDELs were found. The final comparison of sets of SNP positions between the samples was done using BedTools (Quinlan, 2014) and Linux terminal commands (bash).

### Dot blots

DNA was isolated from rho0 cells treated or not with 20 mM DM-$\alpha$-KG for 4 h, denatured in 0.4 M NaOH and 10 mM EDTA at 95°C for 10 min, and neutralized by adding an equal volume of cold 2 M ammonium acetate (pH 7.0). Next, 60 and 30 ng of denatured DNA each were spotted in technical quadruplicates on 0.2-$\mu$m nitrocellulose membranes (Bio-Rad). Blotted membranes were exposed to UV for 10 min, incubated for 2 h at room temperature with 5% bovine serum albumin in 1× PBS with 0.1% Tween 20 (1× PBST) as blocking buffer, and then overnight at 4×C with antibodies raised against 5-meC (rabbit pAB) or 5-hmeC (mouse mAb) (Active Motif). After several washes in 1× PBST, the membranes were incubated with fluorescent secondary antibodies against each host species using different wavelengths (donkey anti-mouse IRDye 680LT and donkey anti-rabbit IRDye 800CW; LiCOR) and visualized using a LiCOR Odyssey imager. DNA templates from four independent biological replicates per group were assayed.

### Data accessibility

Genomics data for this publication have been deposited in the NCBI's Gene Expression Omnibus and are accessible through GEO Series accession number GSE100134.

# Supplementary Information

# Acknowledgements

We thank the staff at the Core Facilities at National Institute of Environmental Health Sciences (NIEHS) and National Institutes of Health (NIH) (Epigenetics and Genomics) and the critical reading of the manuscript by Drs. Matthew Longley and Robert Petrovich (NIEHS). We also thank Dr. Jia-Ji Lin (NIEHS) for help with the hypoxic experiments. This research was supported by the Intramural Research Program of the NIH, NIEHS.

## Author Contributions

OA Lozoya: data curation, formal analysis, validation, investigation, visualization, and methodology.
T Wang: formal analysis.
D Grenet: methodology.
TC Wolfgang: methodology.
M Sobhany: methodology.
D Ganini da SIlva: methodology.
G Riadi: formal analysis.
N Chandel: resources.
RP Woychik: conceptualization, funding acquisition, and writing—review, and editing.
JH Santos: conceptualization, supervision, project administration, and writing—original draft, review, and editing.

## Conflict of Interest Statement

The authors declare that they have no conflict of interest.

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
