## [Reviewer comments · Life Science Alliance]

Life Science Alliance

Mitochondrial acetyl-CoA reversibly regulates locus-specific histone acetylation and gene expression

Oswaldo Lozoya, Tianyuan Wang, Dagoberto Grenet, Taylor Wolfgang, Mack Sobhany, Douglas Ganini da Silva, Gonzalo Riadi, Navdeep Chandel, Richard Woychik, and Janine Santos

DOI: <https://doi.org/10.26508/lsa.201800228>

Corresponding author(s): Janine Santos, NIEHS and Richard Woychik, NIEHS

Review Timeline:

Submission Date:	2018-10-29
Editorial Decision:	2018-10-30
Revision Received:	2018-12-28
Editorial Decision:	2019-01-25
Revision Received:	2019-01-28
Accepted:	2019-01-29

Scientific Editor: Andrea Leibfried

Transaction Report:

Please note that the manuscript was previously reviewed at another journal and the reports were taken into account in inviting a revision for publication at *Life Science Alliance* prior to submission to *Life Science Alliance*.

Referee #1 Review

Report for Author:

Lozoya et al. build on recent publications from their group linking mtDNA loss to histone hypoacetylation. In this manuscript, they study progressive and chronic models of mtDNA depletion. The authors report histone hypoacetylation in both models, together with impaired HAT and ACL activity. These processes were reversed by altering the pool of acetyl-CoA. While acetyl-CoA regulation of histone acetylation is certainly not new, the results are of some interest to the field by extending previous observations to chronic mtDNA loss and by describing functional consequences in terms of gene expression.

However, there are some quite serious questions and experimental issues that remain to be answered before this manuscript could be further considered for publication.

Major concerns:

1. The most problematical issue comes up in Fig 6. The overall decrease in H3K9ac and H3K27ac peaks upon DM-aKG treatment is surprising and must be addressed in more detail. AUC might be more informative and functionally more relevant as peak numbers. Moreover, genes upregulated in rho0 cells rescued by DM-aKG show decreased acetylation - this reviewer fails to understand how these findings support the model in which mtDNA damage leads to hypoacetylation that is rescued by aKG.
2. The Westerns showing acetylation changes are not quantitative and mass spectrometry is the standard in the field. The controls are often insufficient to show that an assay is working. For example, using a kit to assay histone acetylation or deacetylation without any controls to show that the assay is working is not an acceptable experiment.
3. The histone acetylation changes of Cluster B genes (Fig 1b) are hard to interpret. It would be informative to see other histone acetylation marks and chromatin accessibility at Cluster B loci to better understand the functional relevance of these changes. Can the authors speculate of what exactly happening at these loci? How many genes are in this group? How many genes are in Cluster A? What functional categories do these gene sets enrich for?
4. How do other histone marks look in the DN-POLG system? In this study, only H3K9ac was assessed by ChIPseq but other lysine residues are also expected to show changes based on previous findings from the same group.
5. Measuring nuclear acetyl-CoA levels might be more informative with respect to histone acetylation and should be assessed specifically.
6. ACL is only one of several metabolic enzymes that have recently been implicated in epigenetic regulation and histone acetylation. For example, ACSS2 or PDH could contribute as well and should be assessed as no changes in levels or enzymatic activity of ACL were observed.
7. The transcriptional changes in the DM-aKG experiment are not particularly impressive. Out of ~3300 genes differentially expressed in the rho0/+ model, only ~600 are rescued by DM-aKG and only ~420 of these show expected directional changes. In light of this and the fact that aKG is known to regulate TET activity, DNA methylation should be assessed. In addition, the heat map in Fig5a should show all DEGs not just the ones that behave as expected.

Minor comments:

1. Some antibodies, assay names, catalogue numbers are missing and should be provided.

2. Only two biological replicates for the ChIPseq. This is lower than the current standards in the field.
3. Fig 2C is rather low quality.
4. Fig5b is very confusing and hard to read.
5. The text contains many grammatical errors and incomplete sentences.

Referee #2 Review

Report for Author:

In the manuscript "Mitochondrial acetyl-CoA reversibly regulates locus-specific histone acetylation and gene expression" Lozoya and colleagues describe a complementary strategy to assess the effect of mitochondrial DNA loss on histone acetylation in an attempt to understand the molecular mechanisms connecting the two processes.

The authors previously reported (Martinez-Reyes et al., 2016) that progressive mitochondrial DNA depletion inhibits cellular proliferation, without affecting cell viability (in a glucose-rich medium). The authors connected this observation to decreased oxygen consumption rate and mitochondrial membrane potential, slower activity of the TCA cycle, a depleted pool of NAD⁺ and decreased levels of acetylation of lysine residues K9 and K27 of histone H3. In that manuscript, the authors showed that restored oxidation of NADH and FADH₂ to NAD and FAD and the resulting reactivation of the TCA cycle are sufficient for recovery of H3K9ac and H3K27ac levels.

In the present manuscript, the authors expand this line of investigation and question how dysfunctional mitochondria regulate the level of acetyl-coA and how this in turn controls the distribution of H3K9 acetylation and its effect on gene expression.

When expressing a dominant negative mutant of DNA polymerase G (DN-POLG, progressive mitochondrial DNA depletion model), the authors observed a progressive decrease of H3K9ac around transcription start sites. This caused several thousands of genes to be differentially expressed. The authors further show that production of a metabolite from the cholesterol synthesis pathway (downregulated in DN-POLG cells) is affected by the loss of mitochondrial DNA, as indicated in gene expression analysis by RNA-seq. The effects induced by the conditional expression of DN-POLG could be reversed by overexpression of NADH dehydrogenase and alternate oxidase, which restore NAD/NADH and FAD/FADH₂ ratios and activate the TCA cycle.

In parallel, the authors showed that a chronic depletion of mitochondrial DNA can be achieved by treating an osteocarcinoma cell line with low doses of EtBr. As in the case of the progressive mitochondrial DNA depletion model this resulted in decreased acetylation of H3K9. Microarray analysis indicated that this treatment caused several thousands of genes to be differentially expressed. To reverse the effects caused by the EtBr-induced mtDNA depletion the authors stimulated TCA activity. Treatment with cell-permeable α -ketoglutarate (α -KG) restored the levels of acetyl-coA, general HAT activity and promoter-specific H3K9 acetylation. α -KG treatment also increased the amount of COX 2 protein (which was found downregulated in the microarray analysis) and its secreted reaction product.

Both, the progressive and the chronic mitochondrial DNA depletion models suggest that there

is a connection between the loss of TCA cycle activity and the expression of nuclear genes controlled by acetylation of H3K9. The later model argues that the connecting metabolite is acetyl-CoA. The work presented in this manuscript is of interest because it provides a putative novel link between mitochondrial function and the transcription of nuclear genes mediated by acetyl-CoA levels.

The authors use arguments from both the DN-POLG and the EtBr models to derive a mechanism of mtDNA loss-driven histone deacetylation. This is based on the assumption that the two separate models presented in this manuscript follow a common molecular mechanism. However, the link between these models is not yet firmly established and several conclusions of the work are not fully supported by the experiments provided. To move the study beyond an incremental advance over the previous and other work, additional experimentation is required.

General comments

1. a-KG was used to feed the TCA cycle and restore its activity in the EtBr mtDNA-depleted cells. The authors report recovery of acetyl-coA and H3K9ac levels as a result of a-KG treatment. If the two models are indeed complementary, the effects of a-KG treatment need to be analyzed in the DN-POLG cells.

2. The authors use a-KG to feed the TCA cycle. They argue that restored TCA activity is responsible for increased cellular acetyl-CoA levels. As a-KG has pleiotropic effect (i.e. is involved in multiple pathways) the authors need to extend their analysis to other cell permeable metabolites (e.g. pyruvate or other) to feed the TCA cycle and to reproduce the effects of a-KG on acetyl-CoA levels. How do the authors explain the ability of a-KG-supplemented TCA to bypass the need for two more NADs and one more FAD molecule to complete the cycle?

3. Besides its role as an intermediate in the TCA cycle, a-KG is also a co-factor for demethylating enzymes. Since H3K9 (and also H3K27) are targets of methylation besides acetylation it needs to be excluded that the observed a-KG effects are not mediated via affecting histone methylation instead of acetylation. The authors report that many genes were found differentially methylated in the rho0 cells after a-KG treatment. Unfortunately, the identity of these genes is not revealed. Can it be excluded that the promoter of the COX2 gene, whose activity was described in Figure 6, is not one of the genes that were hypermethylated in rho0 cells but demethylated after a-KG-treatment? Is the cholesterol synthesis pathway, which was found affected in the progressive mtDNA loss, also affected by the chronic mtDNA depletion model?

4. a-KG serves as a cofactor not only for DNA demethylating enzymes, but also for histone demethylases. Since both H3K9 and H3K27 are known to define heterochromatic regions when methylated, increased demethylation activity might pose these marks for acetylation. The authors report that acetylation of both K9 and K27 is progressively reduced to 40% of the original levels, but not completely removed. This suggests that acetylation may still take place in both mtDNA-depletion models. To rule out that a-KG does not increase K9 and K27 demethylation, which then poses their acetylation, the authors must perform western blot or mass spectrometry to determine K9 and K27 methylation (and acetylation) over the duration of the a-KG treatment of rho0 (and DN-POLG) cells.

5. The authors report some conflicting findings/statements in this and the previous manuscript with respect to the two models of DNA-depletion. First, the authors do not provide any evidence with respect to the levels of acetyl-coA in the DN-POLG cells as a response to the NAD1/AOX rescue. Second, the authors report increased HAT activity in a-KG treated cells, but no such effect in NAD1/AOX rescued DN-POLG cells. Third, the NAD/NADH ratio was decreased in the DN-POLG model (and can be rescued by NAD1/AOX overexpression), but it seems not to be affected (or was slightly increased) in the EtBr treated cells. These points must be clarified.

Minor comments

1. Western blot analysis must be provided for the DN-POLG (and rescue) time-course experiments in addition to the ChIP analysis. Since the same antibody is being used for detection, the signal should be comparable. Such analysis is required to allow estimating the global levels of the two modifications.
2. There is an extensive technical section detailing the analysis of sequencing data inserted between lines 55-65. This limits the readability of the paper. This part of the manuscript should be moved to the methods section.
3. The authors present evidence that overexpression of NAD1/AOX in the DN-POLG cells recovers the expression of lathosterol (Figure 2F). Since the fold change between the two conditions is incremental, statistical analysis of the data (error bars) is required. Since cells might also exhibit an intrinsic variation in the production of lathosterol, a d0/d9 condition where DN-POLG is not expressed needs to be analyzed in this context.
4. The section describing the tissue culture set up in the Methods is confusing. There is reference to the previously established cell models and then description of new TC systems. However, it is unclear whether the different cells were cultivated the same (aka all cells as written) or differently. This detail has a major impact on any data interpretation.

Referee #3 Review

Lozoya and colleagues study the role for mitochondrial produced acetyl-CoA in regulating locus specific histone acetylation and gene expression using -omics methods. Their data show that chronic or previously reported acute mtDNA loss using biochemical, pharmacologic, genomic and genetic study approaches causes histone hypoacetylation by reduced HAT activity and results in changes mainly in DEG expression, which could help explain or model mtDNA depletion syndromes from a variety of genetic or environmental causes.

A concern raised prior to formal review was based on perceived modest and incremental advance and muted impact for the field. The authors' provided arguments against this editorial view, which for this reviewer (realizing this is an opinion) are not compelling after evaluating the current submission. The authors have published the main finding of their prior acute loss studies and now add technical, study-design refinements that provide a more detailed picture of that work using more sensitive chromatin based detection techniques and mainly confirmatory additional studies. Published work from several fields (sometimes unreferenced in the current study) including immunology, cancer biology, stem cells, and development have shown linkage between specific TCA cycle metabolites and epigenome modifications

impacting gene expression and cell function. Total or 50% losses in mtDNA levels using EtBr and TFAM hemizygote MEFs range to extreme representations of mtDNA depletion that may have limited utility for understanding physiologic or pathologic fluctuations in mitochondrial influence on the epigenome and cell functions.

Some other issues/comments:

1. The inference that ACL enzyme activity is unaffected by mtDNA loss or replenishment in 143B osteosarcoma cells because of similar enzyme and S455 phosphorylation levels in the two conditions does not appear sufficient to make this claim.
 2. Dimethyl-aKG has effects beyond anapleurotic replenishment of the TCA cycle for acetyl-CoA manufacture that were not completely excluded for influencing DEG expression changes in rho⁰ or rho⁺ 143B cells. DM-aKG affects Jumonji class histone demethylases, TET hydroxymethylases (which was examined), and HIF1a stability and enzymatic activity through effects on the prolyl-hydroxylases, all of which could alter epigenome status and DEG expression patterns.
 3. There is no insight provided for how changes in metabolite levels through a variety of manipulations related to the TCA cycle or supplementation by DM-aKG or provision of metabolic enzyme inhibitors affects only specific gene expression. What leads to this specificity? Progress in this question would represent a key advance that the field currently lacks that is related to the cell models and systems developed in this study.
-

October 30, 2018

Re: Life Science Alliance manuscript #LSA-2018-00228-T

Dr. Janine Santos
NIEHS
111 TW Alexander drive
Durham, NC 27709

Dear Dr. Santos,

Thank you for transferring your manuscript entitled "Mitochondrial acetyl-CoA reversibly regulates locus-specific histone acetylation and gene expression" to Life Science Alliance. The manuscript was assessed by expert reviewers at another journal before, and the editor transferred those reports to us with your permission.

The reviewers who evaluated your study elsewhere noted that your work extends previous studies, but that further analyses are needed to better support your conclusions and to provide insight of significant value to others. Based on these reports already at hand, we would like to invite you to submit a revised version for publication in Life Science Alliance. Importantly, we would expect:

- a point-by-point response to the concerns raised
- that the revised version addresses the discrepancies noted by the reviewers
- addition of the requested controls
- improved rescue experiments
- discussion of alternative effects of alpha-KG
- testing the effect on K9 K27 methylation

Such a revised version would undergo technical re-review. I'd be happy to discuss the individual points further with you.

Thank you for this interesting contribution to Life Science Alliance. We are looking forward to receiving your revised manuscript.

Sincerely,

- A letter addressing the reviewers' comments point by point.
- An editable version of the final text (.DOC or .DOCX) is needed for copyediting (no PDFs).
- High-resolution figure, supplementary figure and video files uploaded as individual files: See our detailed guidelines for preparing your production-ready images, <http://life-science-alliance.org/authorguide>
- Summary blurb (enter in submission system): A short text summarizing in a single sentence the study (max. 200 characters including spaces). This text is used in conjunction with the titles of papers, hence should be informative and complementary to the title and running title. It should describe the context and significance of the findings for a general readership; it should be written in the present tense and refer to the work in the third person. Author names should not be mentioned.

B. MANUSCRIPT ORGANIZATION AND FORMATTING:

Full guidelines are available on our Instructions for Authors page, <http://life-science-alliance.org/authorguide>

Referee #1:

Lozoya et al. build on recent publications from their group linking mtDNA loss to histone hypoacetylation. In this manuscript, they study progressive and chronic models of mtDNA depletion. The authors report histone hypoacetylation in both models, together with impaired HAT and ACL activity. These processes were reversed by altering the pool of acetyl-CoA. While acetyl-CoA regulation of histone acetylation is certainly not new, the results are of some interest to the field by extending previous observations to chronic mtDNA loss and by describing functional consequences in terms of gene expression.

However, there are some quite serious questions and experimental issues that remain to be answered before this manuscript could be further considered for publication.

Major concerns:

1. The most problematical issue comes up in Fig 6. The overall decrease in H3K9ac and H3K27ac peaks upon DM-aKG treatment is surprising and must be addressed in more detail. We agree with the reviewer that the decrease in peak number was surprising. However, epigenetic modifications of histones are combinatorial and while for some lysine residues they are exclusive (H3K4, for instance, is exclusively methylated) for other lysines, including H3K9 or 27, they are not. Thus, one explanation for the decrease in peak numbers by DM-aKG is that another post-translational modification became more abundant in these residues in some areas of the genome. Consistent with this, the new data probing methylation of these same residues by Western blots indicate an increase in the levels of H3K9me3 by this treatment. Furthermore, histone succinylation driven by nuclear alpha-ketoglutarate dehydrogenase (OGDH) was recently reported. Although not very abundant because of the low levels of nuclear OGDH, this modification can compete with acetylation of histones (Lu et al., 2017 Nature). It is possible that addition of DM-aKG could have increased this type of modification in the nucleus. We have included information in the manuscript reflecting this point (**page 14 lines 262-266**).

“AUC might be more informative and functionally more relevant as peak numbers”. The reviewer raises an interesting point regarding AUC as an alternative metric for peak calling. Although this metric is not the gold standard in the field, having been shown to perform equally or less efficiently than the methods used in this study (<https://www.ncbi.nlm.nih.gov/pmc/articles/PMC4905608/>), as per the reviewer’s suggestion we applied it to H3K9ac and H3K27ac peaks at genome-wide promoters, and found data that are in agreement with the peak numbers. A **new panel** with this information has now been included in **Fig. S7A**. The functional relevance of peak width based on AUC measurements is still unclear, and only a few studies have utilized this metric to infer functional outcomes in terms of gene expression (<https://www.ncbi.nlm.nih.gov/pubmed/29769529>). Given no differences compared to peak number identified, this was not applied.

“Moreover, genes upregulated in rho0 cells rescued by DM-aKG show decreased acetylation - this reviewer fails to understand how these findings support the model in which mtDNA damage leads to hypoacetylation that is rescued by aKG”. We apologize for the lack of clarity. As stated in the manuscript, while the initial driving force for the current experiments was our previous identification that histones were globally hypoacetylated in the context of mtDNA depletion (Martinez-Reyes et al., 2016) what we found by ChIP-seq in both DN -POLG and the rho0 cells is that there is both loss and gain

of H3K9 or 27ac marks at specific loci as a function of mtDNA loss (see **page 4, lines 35-37**). We understand that data presented in the original Fig. 6B may have added to the confusion as it reflected average changes instead of individual changes at specific loci. We have removed that data and only left Fig. S7B as representing that information. We maintained data showing the peak level on individual loci in the main figures (panels 6C and D).

Despite the bidirectional effect of DM-aKG on peak enrichment, at individual loci the changes in histone acetylation and gene expression were concordant. That is, in loci where the histone modification was decreased in the rho0, DM-aKG led to increase histone acetylation and gene expression while in loci where the modified histone peak decreased by DM-aKG treatment the expression of the gene was downregulated.

2. “The Westerns showing acetylation changes are not quantitative and mass spectrometry is the standard in the field. The controls are often insufficient to show that an assay is working. For example, using a kit to assay histone acetylation or deacetylation without any controls to show that the assay is working is not an acceptable experiment”. Mass spectrometry was performed for the DN-POLG cells and the hypoacetylation of some lysine residues is consistent with the Western blot data (Martinez-Reyes et al., 2016) and the ChIP-seq data (this study). While additional mass spectrometry analysis of histone modifications could reveal the full spectrum of changes in the rho0 cells, the goal of using those cells was to interrogate the extent to which previously identified histone hypoacetylation in H3K9 and H3K27 in the DN-POLG cells were: (1) maintained when mtDNA depletion was chronic and (2) whether they were involved in gene expression regulation. We believe the manuscript as is has achieved these goals. Finally, we would like to call attention for various publications, including recent studies from high profile groups in high impact journals, that used Westerns and ChIP-seq without mass spectrometry data to infer changes in histone mark abundance. These include among others: Wellen et al., Science 2009; Zhao et al., Cell Reports, 2016; Lee et al., Genes and Dev 2018; Mentch et al., Cell Metabolism 2015; Huang et al., Mol Cell, 2018; Ye et al., Mol Cell, 2018.

We have not used a kit to assay histone acetylation or deacetylation but perhaps the reviewer is referring to the HAT and HDAC assays, in which overall enzymatic activities were assayed fluorometrically. As control for the HDAC assays, cells were treated with butyrate, a HDAC inhibitor, for 24h. As expected, butyrate treatment inhibited HDAC function in both cell types. We have now included these data in the manuscript **Fig. 3B**.

As for HAT activity, several different controls were included when experiments were performed despite the data not having been presented in the final figure. They included a positive control (nuclear extracts of cells with high HAT activity as provided by the manufacturer) and negative controls in which we left out of the reaction the sample to be tested or acetyl-CoA. These controls worked as expected and were run together with the assayed samples every time the assay was performed; data has now been included in **Fig. 3C**. While ideally one would treat the cells with a known HAT inhibitor in the assays, HAT inhibitors are non-specific and interfere with assay performance through different mechanisms; for a recent paper on this issue please see <https://www.nature.com/articles/s41467-017-01657-3>.

In our experiments, change in HAT function was observed not only when rho0 cells were assayed but also when the levels of acetyl-CoA were manipulated in either rho+ or rho0 cells through different mechanisms – including the inhibition of ATP citrate lyase. All samples were processed contemporaneously, and experiments with the different cell types or pharmacological manipulations

were performed at the same time. If spurious agents were contaminating or interfering with the assay, it is likely that they would affect all samples. Finally, the results from HAT activity were commensurate with the acetylation of histones themselves (**Fig. 6 and Figs. S5B and C**).

3. The histone acetylation changes of Cluster B genes (Fig 1b) are hard to interpret. It would be informative to see other histone acetylation marks and chromatin accessibility at Cluster B loci to better understand the functional relevance of these changes. Can the authors speculate of what exactly happening at these loci? How many genes are in this group? How many genes are in Cluster A? What functional categories do these gene sets enrich for? Sixty-five genes had peaks that follow the pattern of cluster B (**Page 6, line 84**). Given the few number of genes that present that behavior, we do not believe that genome-wide assays for chromatin accessibility or other histone marks are granted. Because the promoter peak still decreased, in theory these genes could be included in the other pool or could be taken out altogether. However, we thought the behavior was noteworthy and deserving of being reported as such. The pathways enriched by those 65 genes **was provided in Table S1**. While we could speculate what that additional peak may mean (for instance, the expression of a new repeat-driven promoter as we described recently Wang et al., 2016), we believe this may be too premature at this point. Regarding cluster A (**Page 7, line 84**), 1,995 genes followed this pattern and IPA analysis **provided in Table S1** shows that the pathways that they enrich for are relevant to the transcriptome effects associated with mtDNA depletion that we previously reported, including cholesterol biosynthesis and putrescine degradation (Lozoya et al., 2018 PLoS Biology).

4. How do other histone marks look in the DN-POLG system? In this study, only H3K9ac was assessed by ChIPseq but other lysine residues are also expected to show changes based on previous findings from the same group. Several other histone marks were found to be changed in the DN-POLG cells both by Western blots and by mass spectrometry (Martinez-Reyes et al., 2016). We agree with the reviewer that, as such, they would likely change genome-wide as observed for H3K9ac. However, assessing by ChIP-seq how the entire histone modifications landscape change in the DN-POLG is beyond the scope of this manuscript. The aim of the ChIP-seq experiments was to determine the extent to which the previously reported changes in histone acetylation (Martinez-Reyes et al., 2016) played a role in regulation of gene expression. While H3K9ac is certainly not the only histone modification involved in the control of gene expression, the data obtained with the current experiments are consistent with their role in influencing transcriptional output in a locus-specific manner.

5. Measuring nuclear acetyl-CoA levels might be more informative with respect to histone acetylation and should be assessed specifically. We understand the point of the reviewer. However, given that we show that mitochondrial-dysfunction caused by mtDNA depletion influences histone acetylation, that the nuclear/cytosolic pool of acetyl-CoA regulates histone acetylation and that acetyl-CoA freely diffuses between nucleus and cytoplasm (Pietrocola et al., 2015 Cell Metabolism), it is unclear whether attempting to quantify the different pools would change the conclusions of the study. Importantly, there are no standard methods to accurately quantify nuclear vs cytoplasmic pools (Schvartzman et al., 2018 JCB 217:2247–2259).

Finally, because we do not yet know which exact proteins are responsible for the histone effects, how much they potentially traffic between nucleus and cytoplasm, whether they are differentially acetylated and if they interact at the chromatin with proteins (and which proteins, and whether they were differentially acetylated and where), interpreting data on nuclear pools to make inferences on histone acetylation under our experimental conditions will not be trivial. Thus, while we agree that this approach can be valuable, we believe it is beyond the scope of the current manuscript. For a recent review on the limitations of these specific issues, including the challenges of measuring individual metabolic pools, please see Schwartzman et al., 2018 JCB 217:2247–2259.

6. ACL is only one of several metabolic enzymes that have recently been implicated in epigenetic regulation and histone acetylation. For example, ACSS2 or PDH could contribute as well and should be assessed as no changes in levels or enzymatic activity of ACL were observed. We agree with the reviewer and thank for the suggestion; we should mention that we had not initially investigated their role because no changes in the transcript levels of ACSS2 or PDH in either the DN-POLG cells (Lozoya et al., 2018 PLoS Biology) or the rho0 were identified (**Table S3**). Following the suggestion, we have performed Western blots comparing the levels of these proteins in rho+ vs rho0, also looking into their enrichment in the cytosol vs nucleus. Both the pyruvate dehydrogenase complex (PDC) and ACSS2 have been reported to translocate to the nucleus under specific stimuli, such as starvation or hypoxia, to generate a local pool of acetyl-CoA to acetylate histones (Bulusu et al. 2017, Cell Reports; Li et al., 2017 Mol Cell; Mews et al., 2017 Nature). Under our experimental conditions, no changes in the protein amounts were identified by Western blots when comparing rho+ and rho0, we also could not detect ACSS2 or PDH in the nucleus of the cells (**new figure 3F**). The potential contribution of PDH activity was assayed by pharmacological means. We had data that DCA, an inhibitor of the PDKs that in turn increase PDH activity, restored HAT activity in rho0 cells (**Fig. 4C**), indicating that PDH activity is not impaired in the cells. We now added data in which we directly stimulated PDH activity using lipoic acid and found that this too restored HAT activity in rho0 to levels similar to rho+ (**new Fig. S3G**). Thus, we conclude that neither PDH or ACSS2 impairments play a role in the histone acetylation phenotype observed in the rho0 cells. **This new information has now been included in Page 15 lines 170-176.**

7. The transcriptional changes in the DM-aKG experiment are not particularly impressive. Out of ~3300 genes differentially expressed in the rho0/+ model, only ~600 are rescued by DM-aKG and only ~420 of these show expected directional changes. In light of this and the fact that aKG is known to regulate TET activity, DNA methylation should be assessed. In addition, the heat map in Fig5a should show all DEGs not just the ones that behave as expected. We respectively disagree with the reviewer that a 20% change of a well-established transcriptional program by 4h of exposure to DM-aKG is not impressive. Irrespective, as we acknowledged on **page 16**, DM-aKG could be affecting the expression of the 596 genes by altering their DNA methylation status in addition to (or instead of) through histone acetylation. We recently reported that the nuclear DNA of these rho0 cells is mostly hypermethylated, an effect that results from increased DNMT activity and levels of 5meC. Specifically, we showed that the levels of the byproduct of TET function, 5hmeC, were not changed in rho0 compared to rho+. We also reported the 621 DEGs in the rho0 cells that have differential nuclear DNA methylation at their promoter regions relative to the rho+ controls (Lozoya et al., 2018). Thus, if DM-aKG treatment increased activity of TETs, then one would expect that a portion of the loci differentially methylated in the rho0 would be sensitive to DM-aKG. Out of the 596 DEGs affected by the DM-aKG exposure, only

101 were differentially methylated in the rho0 (**Table S5**). Out of these 101, 61 were hypermethylated in the rho0 before the DM-aKG and only 39 of those genes had their expression reversed by the DM-aKG (**Table S5**). Thus, as concluded in the paper, if TET activity was affected by the DM-aKG treatment, only a small fraction (39/596) could have had their expression affected by this enzymatic activity.

Despite this previous information, we went further as per the request of the reviewer and estimated the levels of 5hmeC, the byproduct of TET activity, in the rho0 before and after DM-aKG treatment using dotblots. No differences in the levels of 5hmeC in total genomic DNA were observed. **These new data have now been included as Fig. S8C.**

Minor comments:

- 1. Some antibodies, assay names, catalogue numbers are missing and should be provided.** Included as per request also following journal guidelines.
- 2. Only two biological replicates for the ChIPseq. This is lower than the current standards in the field.** The ENCODE consortium, which provides the guidelines for genomics experiments, recommends at least N=2 for ChIP-seq experiments considering the high cost of such assays. For the most recent guidelines, see (https://www.encodeproject.org/documents/ceb172ef-7474-4cd6-bfd2-5e8e6e38592e/@@download/attachment/ChIP-seq_ENCODE3_v3.0.pdf). Thus, we are following the recommended parameters in the field.
- 3. Fig 2C is rather low quality.** A figure with higher quality is now provided.
- 4. Fig6b is very confusing and hard to read.** As requested, this was removed.
- 5. The text contains many grammatical errors and incomplete sentences.** We have revised the text accordingly.

Referee #2:

In the manuscript "Mitochondrial acetyl-CoA reversibly regulates locus-specific histone acetylation and gene expression" Lozoya and colleagues describe a complementary strategy to assess the effect of mitochondrial DNA loss on histone acetylation in an attempt to understand the molecular mechanisms connecting the two processes.

The authors previously reported (Martinez-Reyes et al., 2016) that progressive mitochondrial DNA depletion inhibits cellular proliferation, without affecting cell viability (in a glucose-rich medium). The authors connected this observation to decreased oxygen consumption rate and mitochondrial membrane potential, slower activity of the TCA cycle, a depleted pool of NAD⁺ and decreased levels of acetylation of lysine residues K9 and K27 of histone H3. In that manuscript, the authors showed that restored oxidation of NADH and FADH₂ to NAD and FAD and the resulting reactivation of the TCA cycle are sufficient for recovery of H3K9ac and H3K27ac levels.

In the present manuscript, the authors expand this line of investigation and question how dysfunctional mitochondria regulate the level of acetyl-coA and how this in turn controls the distribution of H3K9 acetylation and its effect on gene expression.

When expressing a dominant negative mutant of DNA polymerase G (DN-POLG, progressive

mitochondrial DNA depletion model), the authors observed a progressive decrease of H3K9ac around transcription start sites. This caused several thousands of genes to be differentially expressed. The authors further show that production of a metabolite from the cholesterol synthesis pathway (downregulated in DN-POLG cells) is affected by the loss of mitochondrial DNA, as indicated in gene expression analysis by RNA-seq. The effects induced by the conditional expression of DN-POLG could be reversed by overexpression of NADH dehydrogenase and alternate oxidase, which restore NAD/NADH and FAD/FADH₂ ratios and activate the TCA cycle.

In parallel, the authors showed that a chronic depletion of mitochondrial DNA can be achieved by treating an osteocarcinoma cell line with low doses of EtBr. As in the case of the progressive mitochondrial DNA depletion model this resulted in decreased acetylation of H3K9. Microarray analysis indicated that this treatment caused several thousands of genes to be differentially expressed. To reverse the effects caused by the EtBr-induced mtDNA depletion the authors stimulated TCA activity. Treatment with cell-permeable α -ketoglutarate (α -KG) restored the levels of acetyl-coA, general HAT activity and promoter-specific H3K9 acetylation. α -KG treatment also increased the amount of COX 2 protein (which was found downregulated in the microarray analysis) and its secreted reaction product.

Both, the progressive and the chronic mitochondrial DNA depletion models suggest that there is a connection between the loss of TCA cycle activity and the expression of nuclear genes controlled by acetylation of H3K9. The later model argues that the connecting metabolite is acetyl-CoA. The work presented in this manuscript is of interest because it provides a putative novel link between mitochondrial function and the transcription of nuclear genes mediated by acetyl-CoA levels.

The authors use arguments from both the DN-POLG and the EtBr models to derive a mechanism of mtDNA loss-driven histone deacetylation. This is based on the assumption that the two separate models presented in this manuscript follow a common molecular mechanism. However, the link between these models is not yet firmly established and several conclusions of the work are not fully supported by the experiments provided. To move the study beyond an incremental advance over the previous and other work, additional experimentation is required.

General comments

1. α -KG was used to feed the TCA cycle and restore its activity in the EtBr mtDNA-depleted cells. The authors report recovery of acetyl-coA and H3K9ac levels as a result of α -KG treatment. If the two models are indeed complementary, the effects of α -KG treatment need to be analyzed in the DN-POLG cells. Following the reviewer's request, we have treated with DN-POLG cells at day 9 with DM- α KG using the same experimental conditions as for the rho0 cells and have analyzed the final endpoint – histone acetylation. Data demonstrating increased in H3Kac abundance by this treatment in the DN-POLG cells has now been included in the **new Fig. 6C**.

2. The authors use α -KG to feed the TCA cycle. They argue that restored TCA activity is responsible for

increased cellular acetyl-CoA levels. As a-KG has pleiotropic effect (i.e. is involved in multiple pathways) the authors need to extend their analysis to other cell permeable metabolites (e.g. pyruvate or other) to feed the TCA cycle and to reproduce the effects of a-KG on acetyl-CoA levels. The reviewer is correct. We would like to highlight two points that we believe help address the concern of the reviewer. We originally showed that increased activation of pyruvate dehydrogenase (PDH) by DCA, which inhibits the pyruvate dehydrogenase kinases (PDKs), promotes HAT activity and histone acetylation in rho0. Because these cells have impaired mitochondrial function, the growth medium is already supplemented with 100 mM pyruvate to support glycolysis. That this pyruvate contributes to acetyl-CoA levels upon DCA exposure is now supported by new data showing that acetyl-CoA levels are increased by DCA treatment (**new. Fig. S3F**). We also performed HAT activity using lipoic acid, which directly activates PDH, and show that it increases HAT function in the rho0 (**new Fig. S3G**). We hope the reviewer will agree that this evidence supports our conclusions.

How do the authors explain the ability of a-KG-supplemented TCA to bypass the need for two more NADs and one more FAD molecule to complete the cycle? The reviewer raises a very important point. Reducing equivalents are not only generated in the mitochondria through the TCA but also through several other reactions, including in the cytosol. The organelle while not able to directly import/export these species, can use amino acid shuttles to carry the electrons across the inner mitochondrial membrane to replenish the NADH, NAD, NADPH, FAD etc pools such that redox reactions are still maintained in the absence of a functional organelle (or when cancer cells, for instance, switch to anaerobic metabolism). Thus, we believe that it is the activity of these co-transporters (shuttles) that is providing the reducing equivalents to keep the TCA cycle running in the rho0 cells. Consistent with this idea, genes for malate dehydrogenase and the mitochondrial oxoglutarate carrier, the main proteins involved in the malate/aspartate shuttle that carries NADH electrons from the cytosol to mitochondria, are upregulated in the rho0 compared to the rho+ (Table S3).

3. Besides its role as an intermediate in the TCA cycle, a-KG is also a co-factor for demethylating enzymes. Since H3K9 (and also H3K27) are targets of methylation besides acetylation it needs to be excluded that the observed a-KG effects are not mediated via affecting histone methylation instead of acetylation. We thank the reviewer for the suggestion. We have now included data in the rho0 and in the DN-POLG cells showing that DM-a-KG does not affect methylation of H3K9 or H3K27 under our experimental conditions. We also estimated levels of 5hmcC, the byproduct of the TET enzymes, whose activity could also be potentiated by the treatment. No changes in 5hmcC levels were observed. **New data showing lack of changes in H3K9me3, H3K27me3 and 5hmcC levels are now shown in Fig. S8A-C.**

The authors report that many genes were found differentially methylated in the rho0 cells after a-KG treatment. Unfortunately, the identity of these genes is not revealed. We apologize to the reviewer that it was not clear that these data were published in a recent manuscript from our group (Lozoya et al., 2018) to which the reader is referred in the text. However, **Table S5** provided the list of the genes that were sensitive to DM-a-KG treatment and that were also differentially methylated between rho0 and rho+.

Can it be excluded that the promoter of the COX2 gene, whose activity was described in Figure 6, is not one of the genes that were hypermethylated in rho0 cells but demethylated after aKG-treatment? The data presented in our recent work (Table S6 in Lozoya et al. 2018) and in **Table S5** of the current

manuscript reveal that the promoter of PTGS2 is not differentially methylated in the rho0 cells, thus excluding that the treatment led to its demethylation and concomitant increase in gene expression.

Is the cholesterol synthesis pathway, which was found affected in the progressive mtDNA loss, also affected by the chronic mtDNA depletion model? The cholesterol biosynthetic pathway *per se* is not affected in the chronic mtDNA depletion model. However, stearate and some genes involved in triacylglycerol biosynthesis, which are lipid-associated pathways also dependent on acetyl-CoA, are inhibited in the 143B rho0 cells. We have now included in the paper, **revised Fig. 5A**, the pathways enriched by the differentially expressed genes in the rho0 vs the rho+ cells.

4. a-KG serves as a cofactor not only for DNA demethylating enzymes, but also for histone demethylases. Since both H3K9 and H3K27 are known to define heterochromatic regions when methylated, increased demethylation activity might poise these marks for acetylation. The authors report that acetylation of both K9 and K27 is progressively reduced to 40% of the original levels, but not completely removed. This suggests that acetylation may still take place in both mtDNA-depletion models. To rule out that a-KG does not increase K9 and K27 demethylation, which then poises their acetylation, the authors must perform western blot or mass spectrometry to determine K9 and K27 methylation (and acetylation) over the duration of the a-KG treatment of rho0 (and DN-POLG) cells. This was an excellent suggestion by the reviewer. We now include data in the revised manuscript showing that histone methylation is not affected in either cell type upon DM-a-KG exposure cells. New Figs. S8A and B.

5. The authors report some conflicting findings/statements in this and the previous manuscript with respect to the two models of DNA-depletion. First, the authors do not provide any evidence with respect to the levels of acetyl-coA in the DN-POLG cells as a response to the NAD1/AOX rescue. Second, the authors report increased HAT activity in a-KG treated cells, but no such effect in NAD1/AOX rescued DN-POLG cells. We understand the concern of the reviewer from what seemingly is contradictory. The data on citrate and acetyl-CoA levels in the DN-POLG cells was previously published (Martinez-Reyes et al., 2016) but we append it

here as graphical representation, so the reviewer can easily access it. As can be seen, while the average levels of both citrate and acetyl-CoA were higher at day 9 in the NDI1/AOX cells compared to those levels at the DN-POLG also at day 9, the changes are not dramatic and may explain the inability of this levels of acetyl-CoA fluctuation to impact HAT activity when assayed *in vitro*. It is also noteworthy that the time-frame of the DN-POLG and 143B experiments was very different, with mtDNA depletion in the former being assayed in a period of 9 days while the 143B rho0 was generated decades ago. Thus, it is likely that the chronic condition of mtDNA depletion in the rho0 cells led to different adaptation than the one observed in the DN-POLG model,

possibly changing the threshold in which enzymes/proteins that require acetyl-CoA can function. As such, it is likely that they differ in their sensitivity to fluctuations on the levels of acetyl-CoA. The gene expression between the two model systems supports the notion that the long-term mtDNA led to adaptive responses that are in part different than that observed when mtDNA depletion is acute.

Third, the NAD⁺/NADH ratio was decreased in the DN-POLG model (and can be rescued by NAD1/AOX overexpression), but it seems not to be affected (or was slightly increased) in the EtBr treated cells. These points must be clarified. The NAD⁺/NADH ratio was not statistically different in the DN-POLG cells between day 0 and day 9 (Fig. 4G of the Molecular Cell, reproduced here) and follow the same trend as the rho0 cells (right graph). While the ratio was bigger with the rescue at day 9 compared to DN-POLG at day 9, within the NDI1/AOX cells it did not change over the course of the experiment (between day 0 and day 9). We did not access the NAD⁺/NADH ratio in rho0 cells treated with DM-a-KG or any other pharmacological manipulation. As such, the only comparison that can be made is between the acute (DN-POLG) vs the chronic (143B) rho0 cells, which show consistent results.

Minor comments

- 1. Western blot analysis must be provided for the DN-POLG (and rescue) time-course experiments in addition to the ChIP analysis. Since the same antibody is being used for detection, the signal should be comparable. Such analysis is required to allow estimating the global levels of the two modifications.** Western blots in the DN-POLG and the NDI1/AOX cells was published in Martinez-Reyes et al., 2016 Mol Cell. The exact same antibodies were used for the Western blots and for the ChIP-seq experiments in this manuscript. Thus, we disagree that re-doing both Western blots and ChIP-seq experiments to compare efficiency of the same antibodies in the different assays is granted.
- 2. There is an extensive technical section detailing the analysis of sequencing data inserted between lines 55-65. This limits the readability of the paper. This part of the manuscript should be moved to the methods section.** This part has been moved as requested by the reviewer.
- 3. The authors present evidence that overexpression of NAD1/AOX in the DN-POLG cells recovers the expression of lathosterol (Figure 2F). Since the fold change between the two conditions is**

incremental, statistical analysis of the data (error bars) is required. Since cells might also exhibit an intrinsic variation in the production of lathosterol, a d0/d9 condition where DN-POLG is not expressed needs to be analyzed in this context. Data and analyses as requested by the reviewer has now been provided.

4. The section describing the tissue culture set up in the Methods is confusing. There is reference to the previously established cell models and then description of new TC systems. However, it is unclear whether the different cells were cultivated the same (aka all cells as written) or differently. This detail has a major impact on any data interpretation. As requested, this section has been revised.

Referee #3:

Lozoya and colleagues study the role for mitochondrial produced acetyl-CoA in regulating locus specific histone acetylation and gene expression using -omics methods. Their data show that chronic or previously reported acute mtDNA loss using biochemical, pharmacologic, genomic and genetic study approaches causes histone hypoacetylation by reduced HAT activity and results in changes mainly in DEG expression, which could help explain or model mtDNA depletion syndromes from a variety of genetic or environmental causes.

A concern raised prior to formal review was based on perceived modest and incremental advance and muted impact for the field. The authors' provided arguments against this editorial view, which for this reviewer (realizing this is an opinion) are not compelling after evaluating the current submission. The authors have published the main finding of their prior acute loss studies and now add technical, study-design refinements that provide a more detailed picture of that work using more sensitive chromatin based detection techniques and mainly confirmatory additional studies. Published work from several fields (sometimes unreferenced in the current study) including immunology, cancer biology, stem cells, and development have shown linkage between specific TCA cycle metabolites and epigenome modifications impacting gene expression and cell function. Total or 50% losses in mtDNA levels using EtBr and TFAM hemizygote MEFs range to extreme representations of mtDNA depletion that may have limited utility for understanding physiologic or pathologic fluctuations in mitochondrial influence on the epigenome and cell functions. While we agree with the reviewer that total loss of mtDNA has limited physiological meaning as this is not compatible with life, these types of experiments are proof of concept and are in line with an extensive body of literature using KO or even overexpression systems, including in the mouse, which are also non-physiological yet allow understanding of molecular mechanisms. In the DN-POLG model, the effects on the epigenome and cell functions are observed prior to total loss of mtDNA (at days 3 and 6, when mtDNA is depleted but not totally lost); similarly, there is partial loss of mtDNA when EtBr was freshly used (**Fig. S3A**) and in the TFAM cells derived from the heterozygote animals. The degree of depletion of mtDNA in these cells is commensurate with the mtDNA loss observed in patients of the several mtDNA depletion syndromes presenting with clinical symptoms as well as in subject treated with HIV inhibitors. For reviews and example of data in humans, please see <https://www.sciencedirect.com/science/article/pii/S0925443909001410#aep-section-id10>, <https://www.ncbi.nlm.nih.gov/pmc/articles/PMC3985578/> and <https://www.ncbi.nlm.nih.gov/pubmed/12799554>. Thus, we believe that the findings in the current manuscript set the stage to better explore such effects under pathophysiological conditions where mtDNA depletion does occur.

Some other issues/comments:

1. The inference that ACL enzyme activity is unaffected by mtDNA loss or replenishment in 143B osteosarcoma cells because of similar enzyme and S455 phosphorylation levels in the two conditions does not appear sufficient to make this claim. We disagree with the reviewer as S455 phosphorylation has been used as the standard in the field based on earlier data that phosphorylation of S455 increases the enzyme's Vmax and converts its citrate dependence from sigmoidal with negative cooperativity to hyperbolic (Potapova et al., 2000). Furthermore, the production of acetyl-CoA by ACL was also shown to be regulated by this phosphorylation event (Berwick et al., 2002). Finally, phosphorylation of ACL at S455 continues to be used by groups that focus on ACL as proxy for its activity, including Craig Thompson's (Bauer et al., 2005; Wellen et al., 2009 Science) and Kathryn Wellen's (Lee et al., 2014 Cell Metab; Sivanand et al., 2017 Mol Cell.).

2. Dimethyl-aKG has effects beyond anapleurotic replenishment of the TCA cycle for acetyl-CoA manufacture that were not completely excluded for influencing DEG expression changes in rho0 or rho+ 143B cells. DM-aKG affects Jumonji class histone demethylases, TET hydroxymethylases (which was examined), and HIF1a stability and enzymatic activity through effects on the prolyl-hydroxylases, all of which could alter epigenome status and DEG expression patterns. See response to reviewers 1 and 2 above. As per request, levels of HIF1a were analyzed in the samples and were found to be slightly increased after DM-aKG exposure although to lesser extent of when cells were put in hypoxia (**Fig. S6E**). Despite these findings, we found that only 15 genes responding to DM-aKG are known HIF1a targets (**now included in Table S4**). Thus, while HIF1 may contribute to some of the transcriptional response observed under our experimental conditions, its effects are minor (~2% of the genes). This has now been included in the manuscript (**page 13 lines 245-249**).

3. There is no insight provided for how changes in metabolite levels through a variety of manipulations related to the TCA cycle or supplementation by DM-aKG or provision of metabolic enzyme inhibitors affects only specific gene expression. What leads to this specificity? Progress in this question would represent a key advance that the field currently lacks that is related to the cell models and systems developed in this study. The reviewer raises an excellent point. We agree that understanding this issue, which is common in studies where metabolic-driven epigenetic changes occur genome-wide, would lead to a breakthrough in the field. A recent review by Dr. Craig Thompson and colleagues specifically addresses the major challenges in this area (<https://www.ncbi.nlm.nih.gov/pubmed/29760106>). Evidently, addressing this question is not trivial and thus we attempted to speculate in the discussion, including the intriguing possibility that the epigenome-wide changes driven by modulation of mitochondrial function may alter 3D chromatin configuration in a way that spatially facilitates a specific set of genes to be differentially expressed (**Page 21 lines 414-418**). In trying to get some insight into the potential spatial chromatin changes driven by the mitochondrial effects on the epigenome, we compared the enrichment of CTCF binding on the DEGs that had a change in their H3K9ac and H3K27ac upon DM- α KG exposure. CTCF is considered a regulator of chromatin architecture by binding to genetic sequences that in turn can cause the chromatin to loop. Such events are thought to influence gene expression (<https://www.ncbi.nlm.nih.gov/pubmed/30367165>). Our analysis revealed that between 40-70% of the DEGs responding to this treatment also had predicted binding sites for CTCF. While this is encouraging, more in-depth bioinformatics analysis is required. In addition, this hypothesis should be experimentally

tested through omics techniques that can capture chromosome configuration in 3D such as Hi-C, CHIP-loop or ChIA-pet. These experiments are beyond the scope of this manuscript but are being designed.

CTCF bindings sites of the 596 DM-aKG-responsive genes as predicted by HOMER

266 DEGs having CTCF within TSS +/- 10Kb

417 DEGs having CTCF within TSS +/- 25Kb

January 25, 2019

RE: Life Science Alliance Manuscript #LSA-2018-00228-TR

Dr. Janine Santos
NIEHS
111 TW Alexander drive
Durham, NC 27709

Dear Dr. Santos,

Thank you for submitting your revised manuscript entitled "Mitochondrial acetyl-CoA reversibly regulates locus-specific histone acetylation and gene expression". Your work was previously reviewed at another journal and you revised it in response to the concerns raised. We were able to receive input from one of the original reviewers and this reviewer's comments are copied below. We also received arbitrating input from another expert from the field.

As you will see, the reviewer recognizes that the work is improved, but overall remains quite negative. The reviewer thinks that there is no clear demonstration of a mechanistic link between production of mitochondrial acetyl-CoA and locus-specific deacetylation of histone lysine residues; that it remains unclear why aKG increases mito acetyl-CoA levels; and that acetyl-CoA mediated general stimulation of HAT activity argues against specific acetylation/deacetylation events. As mentioned above, we also reached out to an additional advisor. This advisor thinks that while the revised version does not fully address the concerns that were initially raised, the quality and value of the data provided warrant publication in Life Science Alliance.

We think that the original reviewer raises valid points that should get addressed at one point. However, we also appreciate the strength of the current dataset and its resource value to others. We would thus be happy to accept your manuscript for publication here, pending that you address the remaining reviewer concerns by careful discussion and by down-toning your claims. Additionally, the following editorial points should get addressed:

- please provide all figures as a single page file
- please check and add where missing to the figure legends the statistical tests performed to test for significance
- please list 10 authors et al in your reference list
- please add a legend for Fig4F
- please add a callout in the manuscript text for Fig4F, SFig3F
- SFig3H is mentioned in the text but does not exist
- please upload the tables (excel files) you mention in the text

A. FINAL FILES:

-- High-resolution figure, supplementary figure and video files uploaded as individual files: See our detailed guidelines for preparing your production-ready images, <http://life-science-alliance.org/authorguide>

B. MANUSCRIPT ORGANIZATION AND FORMATTING:

Full guidelines are available on our Instructions for Authors page, <http://life-science-alliance.org/authorguide>

Thank you for your attention to these final processing requirements.

Sincerely,

Reviewer #2 (Comments to the Authors (Required)):

The paper has improved, since I first saw it when submitted to another journal. While some of the additional experimental evidence now provided by the authors substantiates their line of argumentation, this is insufficient to clearly demonstrate a mechanistic link between production of mitochondrial acetyl-CoA and locus-specific deacetylation of histone lysine residues.

The authors refuted the possibility that the increased histone acetylation may be a consequence of altered histone and DNA demethylation activities. The authors also convincingly demonstrated that increased pyruvate levels could recapitulate the stimulatory effects of aKG. Finally, the authors showed how aKG treatment of the DN-POLG mutant cell line recovers histone acetylation in the absence of AOX1 and NDI1.

However, it remains unexplained why specifically aKG (and not other TCA cycle metabolites but pyruvate, which can be converted to acetyl-CoA by PDH outside of the TCA cycle) is able to restore the histone acetylation levels in the two mtDNA depletion models.

Moreover, the authors cannot find increased acetyl-CoA levels in the genetic rescue of the DN-POLG mtDNA depletion model. The authors also show that the NADH/NAD ratio, which is expected to increase upon restoration of the electron transport chain by AOX/NDI1 treatment is unaffected by this or by aKG treatment and suggest that reduced NADH₂ and FADH are transported into the mitochondria from the cytosol. This again fails to explain why aKG (and not any other TCA metabolite) increases mitochondrial acetyl CoA levels. Not all observations made in the rescue experiments of the genetic model of mtDNA depletion can thus be reproduced in the Et-Br treated cell line and vice versa.

There is still no clear mechanism to link mitochondrial activity (TCA output or NADH₂/FADH reduction) to acetyl-CoA levels and histone acetyltransferase activity.

Second, a universal mechanism of acetyl-CoA stimulation of HAT activity argues against the specific acetylation/deacetylation of only some lysine residues and of only a specific subset of gene promoters.

The authors must consolidate their findings throughout the results section. The interpretations in the discussion section should be rephrased such that the conclusions inferred from the two mtDNA depletion models converge to a common molecular mechanism that links mitochondrial function to histone acetylation. The findings deduced from one model (TCA activity, acetyl-CoA levels, level of reduced NADH₂/FADH, HAT activity, K9/K27 acetylation) need to be tested in the second model. If this is not possible given the available experimental approaches, the authors must minimally provide a clear resolution of the mechanism which links mitochondrial acetyl-CoA production to histone acetylation. Only then I see the article providing an increase in knowledge over the information already available in the literature.

Reviewer #2 (Comments to the Authors (Required)):

The paper has improved, since I first saw it when submitted to another journal. While some of the additional experimental evidence now provided by the authors substantiates their line of argumentation, this is insufficient to clearly demonstrate a mechanistic link between production of mitochondrial acetyl-CoA and locus-specific deacetylation of histone lysine residues. The authors refuted the possibility that the increased histone acetylation may be a consequence of altered histone and DNA demethylation activities. The authors also convincingly demonstrated that increased pyruvate levels could recapitulate the stimulatory effects of aKG. Finally, the authors showed how aKG treatment of the DN-POLG mutant cell line recovers histone acetylation in the absence of AOX1 and NDI1.

“However, it remains unexplained why specifically aKG (and not other TCA cycle metabolites but pyruvate, which can be converted to acetyl-CoA by PDH outside of the TCA cycle) is able to restore the histone acetylation levels in the two mtDNA depletion models”. We are unsure about this statement from the reviewer as it was not concluded or stated in the manuscript that aKG glutarate is the sole metabolite of the TCA cycle able to restore histone acetylation. In fact, we argue that in either model it is the TCA cycle output that is changed upon mtDNA depletion in turn affecting histone acetylation. While in the DN-POLG model we restarted the cycle by providing NADH oxidation in the electron transport chain, in the chronic model we manipulated different entry points of the TCA cycle which in turn altered its activity. We increased TCA output by pyruvate or a-KG but also decreasing the output in control cells by blocking the mitochondrial citrate carrier or carnitine transporter – both of which caused the opposite effects on histone acetylation. We now have further clarified this in the paper to avoid any misconception for the reader (see Page 10 lines 179-180).

“Moreover, the authors cannot find increased acetyl-CoA levels in the genetic rescue of the DN-POLG mtDNA depletion model”. As provided in the previous response letter, while the levels of acetyl-CoA were not statistically different in the DN-POLG compared to the NDI1/AOX-expressing cells, a trend was identified. The median level of acetyl-CoA in the DN-POLG was 0.5 while in the NDI1/AOX these levels were up to 0.7. Conversely, levels of citrate, the mitochondrial-derived precursor of acetyl-CoA were statistically increased in the NDI1/AOX cells at day 9 compared to the DN-POLG at this same time (Martinez-Reyes et al., 2016).

The authors also show that the NADH/NAD ratio, which is expected to increase upon restoration of the electron transport chain by AOX/NDI1 treatment is unaffected by this or by aKG treatment and suggest that reduced NADH2 and FADH are transported into the mitochondria from the cytosol. This again fails to explain why aKG (and not any other TCA metabolite) increases mitochondrial acetyl CoA levels. Not all observations made in the rescue experiments of the genetic model of mtDNA depletion can thus be reproduced in the Et-Br treated cell line and vice versa. The initial criticism of this reviewer was that there were inconsistent results on the NAD/NADH ratio between the two different models of mtDNA depletion. As provided in the previous response, the ratio of NAD/NADH in the DN-POLG and in the chronic model of mtDNA depletion were similar. Unlike stated by the reviewer, the NAD/NADH ratio was increased in the NDI1/AOX cells when compared to the DN-POLG (Martinez-Reyes et al., 2016). Regarding the aKG exposure of the rho0 cells, we would like to reiterate that we have not measured NADH levels after cells were exposed to aKG. Finally, we would like to highlight that we are not the first ones to show that aKG treatment increases levels of acetyl-CoA. This was previously reported by Guido Kroemer's group in a Molecular Cell paper in 2014 (Marino et al., 2014, which is cited within our manuscript). In fact, this paper was used as the reference for all pharmacological manipulations of the TCA cycle performed here. Their study states that “all treatments that increased total cellular AcCoA

levels (DCA, LA, DMKG, KIC, and UK5099) also raised the cytosolic concentration of AcCoA". In our work we have tested DCA, LA, DMKG and UK5099, all of which affect specific TCA cycle activities and increase the total level of acetyl-CoA. Because their targets are specific mitochondrial enzymes, we conclude that their overall effects in total acetyl-CoA derives from the increased TCA cycle, and thus mitochondrial, output.

"There is still no clear mechanism to link mitochondrial activity (TCA output or NADH2/FADH reduction) to acetyl-CoA levels and histone acetyltransferase activity". As discussed in the manuscript, several hypotheses are raised and need to be further tested to define exactly how the mitochondrial output of acetyl-CoA influences HAT function (**Pages 17-18 lines 338-346**).

"Second, a universal mechanism of acetyl-CoA stimulation of HAT activity argues against the specific acetylation/deacetylation of only some lysine residues and of only a specific subset of gene promoters." As we mentioned in the previous rebuttal letter, the reviewer raises a very important point that has yet to be solved by the field. Metabolite-specific driven changes in the epigenome and gene expression have been previously reported but how specificity can be achieved when changes are genome-wide have yet to be defined. This specific issue was recently raised in an excellent review by Dr. Craig Thompson and colleagues at JCB. Nonetheless, changes in HAT affinity for specific lysine residues as a function of acetyl-CoA concentrations have been previously reported as has been the modulation of activity of specific HATs (like GCN5 or EP300) based on acetyl-CoA fluctuations. Thus, while fundamental to understanding how specificity can be achieved, addressing these questions experimentally are well-beyond the scope of the current manuscript. Instead, they were brought up in the discussion (**Page 19 lines 339-359 and Page 20, lines 391 – 398**).

"The authors must consolidate their findings throughout the results section. The interpretations in the discussion section should be rephrased such that the conclusions inferred from the two mtDNA depletion models converge to a common molecular mechanism that links mitochondrial function to histone acetylation. The findings deduced from one model (TCA activity, acetyl -oA levels, level of reduced NADH2/FADH, HAT activity, K9/K27 acetylation) need to be tested in the second model. If this is not possible given the available experimental approaches, the authors must minimally provide a clear resolution of the mechanism which links mitochondrial acetyl-CoA production to histone acetylation. Only then I see the article providing an increase in knowledge over the information already available in the literature." We respectively disagree with the reviewer. The two model systems and approaches presented while complementary are different in many aspects (for instance, the duration of the mtDNA depleted state) and thus do not warrant consolidation of the findings in one section. Exactly because the approaches and models were complementary they build on each other reinforcing the same conclusion: that the acetyl-CoA output derived from the TCA cycle influences HAT activity and histone acetylation in the nucleus. While the reviewer indicates that we need to repeat all experiments done in one cell model on the other, the reviewer fails to recognize that we provide evidence of decreased TCA cycle output as a function of mtDNA depletion, decreased HAT activity, decreased histone acetylation of both K9 and K27 in both models. We also provide evidence that TCA cycle output strongly correlates with histone acetylation in the nucleus by either increasing NADH oxidation in the DN-POLG or providing both the DN-POLG and the rho0 cells with DM-a-KG. As mentioned above, while no statistical difference for acetyl-CoA levels in the DN-POLG cells were obtained, a trend toward rescue based on NDI1/AOX overexpression was identified. Thus, we feel strongly that our results unequivocally demonstrate that dysfunctional mitochondria, by affecting the TCA cycle, impact the maintenance of the histone acetylation landscape. The common mechanism between the two models is provided in the Discussion (**Page 18 lines 359-375**).

January 29, 2019

RE: Life Science Alliance Manuscript #LSA-2018-00228-TRR

Dr. Janine Santos
NIEHS
111 TW Alexander drive
Durham, NC 27709

Dear Dr. Santos,

Thank you for submitting your Research Article entitled "Mitochondrial acetyl-CoA reversibly regulates locus-specific histone acetylation and gene expression". We appreciate the introduced changes and it is a pleasure to let you know that your manuscript is now accepted for publication in Life Science Alliance. Congratulations on this interesting work.

DISTRIBUTION OF MATERIALS:

Again, congratulations on a very nice paper. I hope you found the review process to be constructive and are pleased with how the manuscript was handled editorially. We look forward to future exciting submissions from your lab.

Sincerely,
